# Filming enhanced ionization in an ultrafast triatomic slingshot

Andrew J. Howard [1,2✉], Mathew Britton[2,3], Zachary L. Streeter[4,5], Chuan Cheng[6], Ruaridh Forbes [2,7], Joshua L. Reynolds [1], Felix Allum [2,7], Gregory A. McCracken[1,2], Ian Gabalski[1,2], Robert R. Lucchese [5], C. William McCurdy[4,5], Thomas Weinacht[6] & Philip H. Bucksbaum [1,2,3,7✉]

Filming atomic motion within molecules is an active pursuit of molecular physics and quantum chemistry. A promising method is laser-induced Coulomb Explosion Imaging (CEI) where a laser pulse rapidly ionizes many electrons from a molecule, causing the remaining ions to undergo Coulomb repulsion. The ion momenta are used to reconstruct the molecular geometry which is tracked over time (i.e., filmed) by ionizing at an adjustable delay with respect to the start of interatomic motion. Results are distorted, however, by ultrafast motion during the ionizing pulse. We studied this effect in water and filmed the rapid "slingshot" motion that enhances ionization and distorts CEI results. Our investigation uncovered both the geometry and mechanism of the enhancement which may inform CEI experiments in many other polyatomic molecules.

[1] Department of Applied Physics, Stanford University, Stanford, CA 94305, USA. [2] Stanford PULSE Institute, SLAC National Accelerator Laboratory, 2575 Sand Hill Road, Menlo Park, CA 94025, USA. [3] Department of Physics, Stanford University, Stanford, CA 94305, USA. [4] Department of Chemistry, University of California, Davis, Davis, CA 95616, USA. [5] Chemical Sciences Division, Lawrence Berkeley National Laboratory, Berkeley, CA 94720, USA. [6] Department of Physics and Astronomy, Stony Brook University, Stony Brook, NY 11794, USA. [7] Linac Coherent Light Source, SLAC National Accelerator Laboratory, Menlo Park, CA 94025, USA. ✉email: ahow@stanford.edu; phbuck@stanford.edu

Hydrogen atoms within molecules can move extremely rapidly in response to the sudden intramolecular forces introduced by ionization or photoexcitation. These atoms can accelerate to traverse Angstrom-scale distances in just tens of femtoseconds. Such ultrafast motion may mediate many biologically important light-matter interactions, including photosynthesis, photochemical damage mitigation in DNA, and vision[1–3]. Resolving the femtosecond-scale motion of these light atomic species is therefore central to the fields of molecular physics and quantum chemistry, and motivates continued efforts to film atomic-scale "molecular movies"[4–6].

Many methods exist to record ultrafast motion in molecules including diffractive imaging techniques, such as ultrafast electron diffraction[7–9], laser-induced electron diffraction[10,11], and hard x-ray diffraction[12–14]; spectroscopic techniques such as high harmonic generation[15–17]; and momentum imaging techniques such as Coulomb Explosion Imaging (CEI)[18–33]. Diffractive techniques, however, lose sensitivity with decreasing atomic mass and spectroscopic techniques typically rely on indirect observables of atomic motion such as electronic structure. Among the methods listed, only CEI probes the direct momenta of all atoms, irrespective of mass, within a molecule[23,31–33,34].

CEI deduces the positions of atoms within molecules by stripping away binding electrons and measuring the momenta of fragments produced in the resulting Coulomb repulsion of the ions. The binding electrons can be removed using thin foils[27,35], intense infrared (IR) lasers[18–21,36] or ultrafast x-ray pulses[22–26,28,37]. Intense IR lasers are especially attractive for time-resolved CEI; intramolecular dynamics can be initiated with a few-femtosecond table-top IR laser pulse and probed with sub-femtosecond temporal jitter using another pulse derived from the same source[38,39].

One of the greatest limitations of laser-induced CEI is that multiple ionization typically occurs sequentially; as a result, intermediate charge states can drive nuclear dynamics prior to Coulomb explosion[40,41]. The ultimate dissociation pathway of the fragments is therefore rarely determined by idealized Coulomb repulsion between point-like ions. This problem is greatly exacerbated by a strong-field phenomenon known as Enhanced Ionization (EI), where a "critical" spacing among the constituent atoms of a molecule increases the ionization yield[42–47], distorting the momentum distribution observed via CEI to favor the critical geometries that undergo EI.

Strong-field distortions of molecular dissociation dynamics were first studied extensively in diatomic molecules[42–44,48–50] and have been more recently studied in triatomic molecules[40,41,45,51–53] such as water[46,54–59]. Diatomics have long been known to undergo EI en route to dication formation: strong-field ionization of the first electron drives nuclear motion toward a geometry that facilitates ionization of the second electron. In this geometry the two atoms are always stretched to a critical distance and aligned with the polarization axis of the laser[60–64]. If both electrons are removed within the same pulse, it has been found that decreasing the pulse duration generally reduces the fraction of molecules that undergo EI[40,41,65]. A similar phenomenon exists in multiply-charged linear triatomics[45,53,66], and has recently been studied both experimentally[46] and theoretically[67] in triply charged water, the bent triatomic considered in this work.

Here, we used $D_2O$ as a model molecular system to film an EI process that proceeds within just 20 fs. To do so, we first verified that the formation of $D_2O^{3+}$ via strong-field multiple ionization yields CEI results with clear distortions indicative of EI. We characterized how the severity of these distortions greatly depends on the ionizing pulse duration. We then demonstrated that the conditions for EI can be reproduced by launching rapid

"slingshot" motion in $D_2O^{2+}$. We modeled this motion using ab initio theory, and the correspondence between theory and experiment allowed the direct retrieval of the time-resolved molecular geometry. The resulting molecular movie revealed the critical geometry at which EI occurs and unveiled the underlying mechanism that induces the enhancement. This improved understanding of EI can not only aid the analysis of future CEI data, but can also be employed to highlight particular features of polyatomic motion in future atomic-scale molecular movies.

## Results and discussion

**Eliciting EI in water.** Two experimental schemes were used to induce EI in water, both of which are depicted in Fig. 1a. In the first scheme, 800 nm pulses of variable duration ($\tau = 6, 10, 19,$ and 40 fs) but constant peak intensity ($I_0 = 2 \times 10^{15}$ W/cm$^2$) ionize neutral $D_2O$ to form $D_2O^{3+}$. Here, the intermediate charge states, $D_2O^+$ and $D_2O^{2+}$, undergo field-assisted dynamics within the envelope of a single pulse. In the second scheme, a 6-fs 750 nm pulse ($I_0 = 1 \times 10^{15}$ W/cm$^2$) doubly ionizes the neutral molecule to create the dication $D_2O^{2+}$ before an identical cross-polarized pulse follows at an adjustable delay ($\Delta t = 10–110$ fs) to form the trication $D_2O^{3+}$. Here, the dication undergoes field-free dynamics in the time between pulses. Following trication formation, the molecule rapidly Coulomb exploded into three bodies ($D^+/D^+/O^+$), each of whose three-dimensional momenta were captured in coincidence by a high-resolution position- and time-sensitive detector[68]. (See Methods for further details on the experimental design.)

Both long pulses and pulse pairs set to a particular interpulse delay were found to more efficiently strip three electrons from $D_2O$ than single short pulses at the same intensity (Fig. 1b and c.) For single pulses, we found that the ratio ($R$) of triply-($D^+/D^+/O^+$) to doubly-charged ($D^+/D^+/O$) three-body dissociations undergoes a 27-fold increase as the pulse duration lengthens from 6 to 19 fs (Fig. 1b). Using pulse pairs, a lower peak intensity was chosen to purposely highlight any enhancement, and consequently $R$ increased nearly 42 times when the interpulse delay was set to 18 fs as compared to a single pulse at the same intensity (Fig. 1c). We now examine the CEI observables in the one-pulse and two-pulse data to find the dynamics responsible for each enhancement.

The fragment momenta captured following ionization with single pulses are strongly suggestive of ultrafast nuclear motion. These momenta were transformed to the molecular frame defined (by coordinates $x_m$, $y_m$, and $z_m$) in the lower left corner of Fig. 1a (see Methods for a mathematical description of this transformation). Figure 2a–d plots the molecular-frame momenta of all three fragments present in every $D^+/D^+$ coincidence following ionization at each of the four pulse durations studied. In Fig. 2a, for example, the higher-momentum $D^+$ cluster results from triple ionization ($D^+/D^+/O^+$), and the lower-momentum $D^+$ cluster results from double ionization ($D^+/D^+/O$)[57–59]. The ratio of these two fragmentation pathways for the four pulse durations is shown in Fig. 1b. (Figure 2d reveals why this ratio ultimately decreases as the pulse duration is increased from 19 to 40 fs: a narrowly distributed $D^+/D^+/O$ feature appears when $\tau = 40$ fs which we attribute to the relatively slow unbending of $D_2O^{+}$[58] facilitating second ionization and subsequent three-body dissociation). As this ratio changes, the momentum distributions within the $D^+/D^+/O^+$ channel also change due to particular geometrical distortions: stretching of the OD bond lengths ($r_{OD}$) reduces the momenta of all fragments, unbending of the DOD bond angle ($\theta_{DOD}$) increases the angle between the two $D^+$ momenta ($\beta$), and alignment sees the angle between the D-D axis and the laser polarization axis ($\theta$) tend toward 0 and 180°.

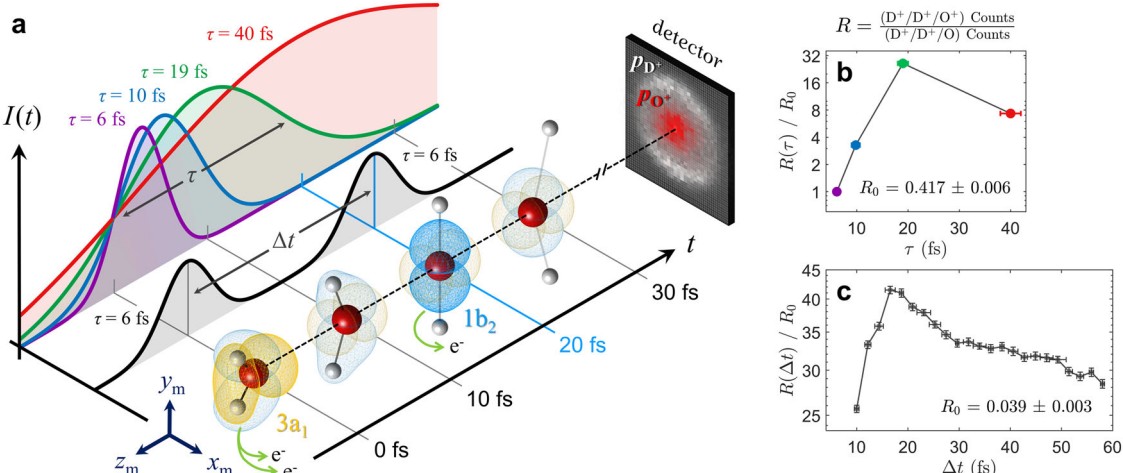

**Fig. 1 Strong-field enhanced ionization forming $D_2O^{3+}$. a** A schematic of the experiment. On the left, the temporal intensity profiles, $I(t)$, are displayed for single pulses of variable duration, $\tau$, (in color) and pulse pairs with variable interpulse delay, $\Delta t$, (in black). Ultrafast nuclear dynamics are initiated by the ionization of two electrons from neutral $D_2O$ at $t = 0$. Upon reaching a critical geometry at $t = 20$ fs, Enhanced Ionization (EI) occurs to facilitate the ionization of a third electron. Following formation of $D_2O^{3+}$, three molecular fragments ($D^+/D^+/O^+$) are produced and mapped to a detector screen, each imprinting a three-dimensional momentum distribution ($p_{D^+}$ and $p_{O^+}$) dependent on the molecular geometry prior to dissociation. Two molecular orbitals, $3a_1$ and $1b_2$, are highlighted as the molecular geometry distorts. **b** The measured ratio ($R$) of triply-charged ($D^+/D^+/O^+$) to doubly-charged ($D^+/D^+/O$) three-body dissociations plotted logarithmically as a function of pulse duration (where $I_0 = 2 \times 10^{15}$ W/cm$^2$) and scaled to the ratio for a 6 fs pulse ($R_0 = 0.417 \pm 0.006$). **c** The same ratio as in (**b**) plotted as a function of interpulse delay (where $I_0 = 1 \times 10^{15}$ W/cm$^2$) and scaled to the ratio for a single 6 fs pulse ($R_0 = 0.039 \pm 0.003$). In both (**b**) and (**c**), EI manifests as $R/R_0 > 1$. Error bars represent plus or minus one standard deviation.

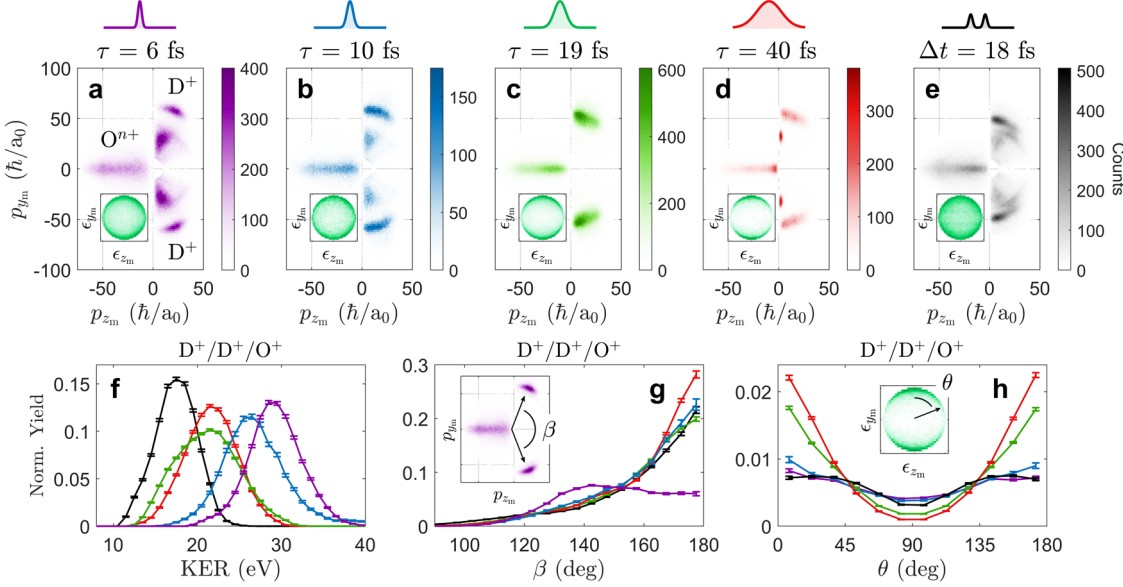

**Fig. 2 Pulse-shape dependence of $D_2O^{2+}$ & $D_2O^{3+}$ fragment momenta. a–e** The molecular-frame momentum distribution (in $p_{z_m}$ and $p_{y_m}$) of the three fragments ($D^+/D^+/O^{n+}$ where $n = 0,1$) present in all $D^+/D^+$ coincidences. In each plot, the $D^+$ momentum distributions appear in the upper and lower right quadrants ($p_{z_m} > 0$) while the $O^{n+}$ momentum distribution appears on the left ($p_{z_m} < 0$). **a–d** Correspond to ionization via single pulses with durations ranging from 6 to 40 fs. **e** Corresponds to ionization via pulse pairs with an interpulse delay of 18 fs. The inset in (**a–e**) displays a crushed 2-dimensional projection (in $\epsilon_{z_m}$ and $\epsilon_{y_m}$) of the normalized 3-dimensional polarization vector ($\hat{\epsilon}$) in the molecular frame. **f–h** One-dimensional distributions of the total kinetic energy release (KER), momentum-frame bend-angle, $\beta = \arccos(\hat{p}_{D^+_{(1)}} \cdot \hat{p}_{D^+_{(2)}})$, and molecular-frame alignment angle, $\theta = \arccos(\hat{y}_m \cdot \hat{\epsilon})$, for each $D^+/D^+/O^+$ distribution displayed in (**a–e**). Error bars represent plus or minus one standard deviation.

Stretching, unbending, and alignment are evidenced by the one-dimensional distributions shown in Fig. 2f, g, and h respectively. Figure 2f shows the total kinetic energy release (KER) decreasing from ~30–20 eV as pulse duration is increased from 6 to 19 fs. This trend suggests that $r_{OD}$ stretches as the pulse duration increases, reducing the KER of the Coulomb explosion, until a specific bond length is reached. Figure 2g shows $\beta$ rapidly increasing from ~140 to 180° as the pulse duration is increased

from 6 to 10 fs, suggesting that the molecule may unbend in <10 fs; however, rapid unbending motion can cause $\beta$ to reach 180° even when $\theta_{DOD}$ is <180°. Finally, Fig. 2h shows that $\theta$ grows more sharply peaked at 0 and 180° as pulse duration increases, suggesting dynamic alignment of the molecule's D–D axis with the laser polarization axis[55,57].

Next we compare the single-pulse momentum distributions to the double-pulse momentum distributions at an interpulse delay

of 18 fs, the delay at which $R$ is maximized (Fig. 1c). Figure 2e shows that ionization by pulse pairs leads to the same three intramolecular distortions as seen with single pulses. 1-dimensional distributions of KER, $\beta$, and $\theta$ for this particular interpulse delay are likewise reproduced in Fig. 2f–h, also showing evidence for stretching, unbending, and alignment correlated with increased trication formation. The relative uniformity of the distribution in $\theta$ for $\Delta t = 18$ fs, as compared to the cases of $\tau = 20$ or 40 fs, however, indicates that dynamic alignment likely plays a much larger role in the nuclear dynamics en route to enhanced ionization when multiply ionizing with single long ($\tau \geq 20$ fs) pulses.

**Recording and modelling intramolecular motion.** Ionizing with pulse pairs and tracking the CEI observables as a function of interpulse delay offers a wealth of information on the intramolecular dynamics leading to enhanced trication formation. When paired with detailed ab initio calculations, these observables serve as an unambiguous probe of state-selective interatomic motion. In our theoretical treatment of the nuclear dynamics, the leading pulse in the pair promotes the Wigner phase space distribution of the ground vibrational state of neutral $D_2O$ to the accurately computed potential energy surface of any one of the nine states of $D_2O^{2+}$ with two vacancies in the valence orbitals[29,30,69,70]. These nine states are detailed in Table 1 of Methods. This phase space distribution is then propagated using classical trajectories on each of the nine potential energy surfaces for a time, $\Delta t$, before arrival of the second pulse and formation of the trication. Upon formation of the trication, the trajectories are continued under simple Coulomb repulsion of three singly-charged fragments, allowing the straightforward extraction of the asymptotic fragment momenta for each delay (see Methods for further detail on this procedure).

Comparing the experimental observables to their simulated counterparts allows us to identify which nuclear dynamics in $D_2O^{2+}$ ultimately contribute to the formation of $D_2O^{3+}$. In Fig. 3a–c, three experimental observables are plotted for all $D^+/D^+/O^+$ coincidences over interpulse delay: the magnitude of the deuteron momentum ($p_{D^+}$) the magnitude of the oxygen-ion momentum ($p_{O^+}$) and momentum-frame bend angle ($\beta$). In Fig. 3d–f, the same three observables are obtained from an ensemble of calculated trajectories and plotted similarly, reproducing many qualitative features of the data (see section 2.1 of the Supplementary Information including Fig. S1 and Table S1 for details).

In order to emphasize the distinct features within Fig. 3d–f, two example trajectories are highlighted, one labeled "slingshot" and the other labeled "two-body". These two trajectories are depicted

in a series of film-strip plots in Fig. 3g–k and Fig. 3l–p, respectively. These plots display the instantaneous molecular geometry and asymptotic fragment momenta for five particular values of $\Delta t$. The first trajectory, labeled "slingshot", is an example of symmetric breakup into three bodies. Here, both deuterons move symmetrically following double ionization and the molecule undergoes a rapid slingshot motion in which the bend-angle is inverted about the $z_m$ axis. As the molecule unbends and stretches, it briefly becomes linear at $\Delta t = 20$ fs before re-bending the other way. This kind of slingshot trajectory only occurs on three of the nine states of $D_2O^{2+}$, and it is most common in the relatively high-lying 2 $^1A_1$ state[29,70] (see Table 1 for branching ratios). The second trajectory, labeled "two-body", is an example of asymmetric breakup. Here, only one deuteron is ejected following double ionization, leaving the other deuteron orbiting the oxygen atom as part of a rotationally and vibrationally hot $OD^+$ fragment. This kind of two-body trajectory occurs on four of the nine states of $D_2O^{2+}$ and is most common in the lowest three states ($^3B_1$, 1 $^1A_1$, and $^1B_1$)[29,70].

Atomic motion is revealed in detail by comparing the data to the two trajectories highlighted in Fig. 3. For example, Fig. 3a displays $p_{D^+}$ decreasing monotonically before bifurcating into two branches at later delays. Each branch is well approximated by a highlighted trajectory in Fig. 3d, representing a deuteron that escaped via the slingshot trajectory or the deuteron that first escaped via the two-body trajectory. In Fig. 3c, $\beta$ is centered at 180° after only 10 fs. Figure 3h and m demonstrate that this corresponds to a $\theta_{DOD}$ of only 146° and 158°, respectively. This discrepancy between $\beta$ and $\theta_{DOD}$ represents a clear example of what is known as nonaxial recoil or a breakdown of the axial recoil approximation[29,70]. In the slingshot trajectory, the molecule becomes linear at $\Delta t = 20$ fs, pictured in Fig. 3i, at which time $\beta$ has already bent backwards, to 148°. The backwards bend also reverses the momentum of the oxygen ion, as seen in Fig. 3g–k. Evidence for this reversal is seen in the earliest delays of Fig. 3b where the measured oxygen momentum appears to pass through zero.

Additional two-body motion can also be revealed by comparing the data with the appropriate trajectory. Figure 3c displays a feature that bends rapidly to $\beta \sim 0°$ and then unbends after about 60 fs. The two-body trajectory in Fig. 3f reproduces this feature. Figure 3l–p demonstrates how the bound deuteron rotates around the oxygen atom within 60 fs such that its momentum again aligns with the dissociating deuteron and yields $\beta = 0°$. Likewise, in Fig. 3b, the faint cluster near $p_{O^+} \sim 80$ $\hbar/a_0$ at $\Delta t \sim 60$ fs can also be attributed to this same motion in the two-body trajectory. Here, both deuterons are oriented to act together in repulsing the oxygen ion, giving it maximal momentum (Fig. 3o).

**Table 1 Electronic states of $D_2O^{2+}$.**

| $\Delta E$ (eV) | $C_{2v}$ Symmetry | $C_s$ Symmetry | Orbital configuration | 2/3-Body branching ratio (%) | Rapid slingshot |
|---|---|---|---|---|---|
| 40.3 | $^3B_1$ | $^3A''$ | $(3a_1)^{-1}(1b_1)^{-1}$ | 93.0/7.0 | Yes |
| 41.4 | 1 $^1A_1$ | 1 $^1A'$ | $(1b_1)^{-2}$ | 99.4/0.6 | No |
| 42.8 | $^1B_1$ | 1 $^1A''$ | $(3a_1)^{-1}(1b_1)^{-1}$ | 87.7/12.3 | Yes |
| 44.3 | $^3A_2$ | 2 $^3A''$ | $(1b_2)^{-1}(1b_1)^{-1}$ | 0.0/100.0 | No |
| 46.0 | $^1A_2$ | 2 $^1A''$ | $(1b_2)^{-1}(1b_1)^{-1}$ | 0.0/100.0 | No |
| 46.0 | 2 $^1A_1$ | 2 $^1A'$ | $(3a_1)^{-2}$ | 26.3/73.7 | Yes |
| 46.3 | $^3B_2$ | 1 $^3A'$ | $(1b_2)^{-1}(3a_1)^{-1}$ | 0.0/100.0 | No |
| 48.4 | $^1B_2$ | 3 $^1A'$ | $(1b_2)^{-1}(3a_1)^{-1}$ | 0.0/100.0 | No |
| 53.6 | 3 $^1A_1$ | 3 $^1A''$ | $(1b_2)^{-2}$ | 0.0/100.0 | No |

A tabulated list of all nine states of $D_2O^{2+}$ that correspond to the removal of two electrons from any combination of the valence orbitals: $(1b_2)^2(3a_1)^2(1b_1)^2$ as labeled by $C_{2v}$ symmetry. For each state, the energy, symmetry, orbital configuration, 2/3-body branching ratio, and existence of rapid slingshot trajectories are listed. Here, the energy is written in terms of $\Delta E$, the energy difference between the neutral ground state of $D_2O$ at the equilibrium geometry and the Franck-Condon point of a given state. These energies come directly from the potential energy surfaces calculated by Gervais et al.[69] and Streeter et al.[70]. The 2/3-body branching ratios were found via simulation after propagating 2048 classical trajectories on each surface. "Rapid slingshot" refers to the whether or not each state permits a 3-body dissociation trajectory that inverts the $z_m$ axis of the molecule within 20 fs.

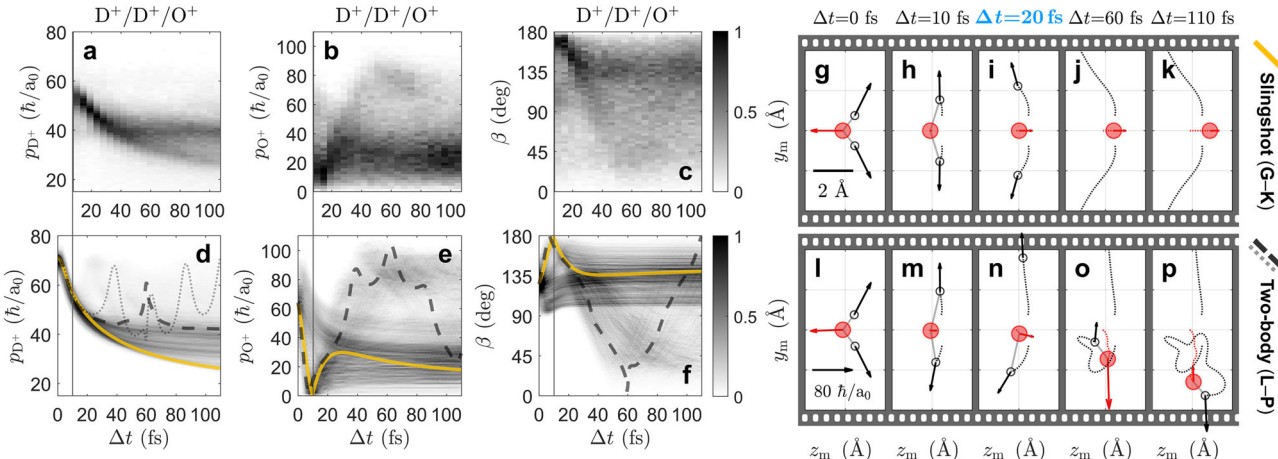

**Fig. 3 Filming nuclear motion in $D_2O^{2+}$. a–c** Measured distributions of deuteron momentum ($p_{D^+}$), oxygen-ion momentum ($p_{O^+}$), and momentum-frame bend-angle ($\beta$) plotted as a function of interpulse delay ($\Delta t$), detected in the triply charged three-body dissociation channel ($D^+/D^+/O^+$). **d–f** Theoretical distributions of $p_{D^+}$, $p_{O^+}$, and $\beta$ as a function of $\Delta t$. **g–k** A series of film-strip plots for the class of trajectory labeled "slingshot". These plots display both the position of the atoms (red solid circles for oxygen and black open circles for hydrogen) and their corresponding asymptotic momenta following formation of the trication. **l–p** Another series of film-strip plots for the class of trajectory labeled "two-body". The slingshot and two-body trajectories are highlighted in (**d–f**) where a solid yellow line represents the slingshot trajectory, and the dashed (and dotted) black lines represent the two-body trajectory.

**Extracting the critical geometry**. We can now determine which geometries are responsible for the enhancement in trication production observed at particular pulse-pair delays. The trication yield in Fig. 1c is maximal for $\Delta t = 18$ fs. Near this particular delay (at $\Delta t = 20$ fs), the slingshot trajectory, for example, traverses through the geometry in which the molecule is briefly linear (Fig. 3i). Here $r_{OD}$ has stretched to ~2.2 Å. To fully explore each distorted molecular geometry and find its contribution to the enhancement, we examined all of the simulated trajectories that comprise Fig. 3d–f.

We extracted the critical geometry of EI from the trajectories by utilizing the fact that the observed enhancement is localized not only in time but in momentum and angle. Using three observables ($\Delta t$, $\beta$, and $p_{D^+}$) we constructed a histogram that localizes the "enhancement volume" in 3-dimensions. An isointensity surface of this 3-dimensional volume at 50% maximum enhancement is depicted in Fig. 4a. To retrieve geometrical information at the moment of the enhancement, we propagated all trajectories through this 3-dimensional space and assigned each trajectory a weight per time-step based on the local value of the enhancement (see section 2.2 of the Supplementary Information including Fig. S2 for details). Two sample trajectories for each state are plotted traversing through this 3-dimensional space in Fig. 4a. The three states that undergo rapid slingshot motion, bending backwards in ≤20 fs, are highlighted in yellow, whereas all other states are depicted in black. Figure 4b shows the results of this analysis, plotting the combined contributions of all weighted trajectories in coordinates of $r_{OD}$ and $\theta_{DOD}$. This reveals an enhancement for bond lengths between 1.8 and 2.5 Å as well as bend angles between 160 and 180°, with the maximum of the distribution occurring at $r_{OD} = 2.2$ Å and $\theta_{DOD} = 180°$.

The inset of Fig. 4b reveals that it is only the states that undergo rapid slingshot motion (highlighted in yellow) that make a large contribution to this enhancement. (Slower slingshot motion occurs more rarely: <1% of dissociations on the $1^1A_1$ state. This state is responsible for the black trajectories that traverse through the enhancement volume, as seen in Fig. 4a and b.) Accessing any one of the states in yellow ($^3B_1$, $^1B_1$, or $2$ $^1A_1$) involves forming at least one vacancy in the $3a_1$ molecular orbital while the $1b_2$ orbital remains doubly occupied (see Fig. 1a for a schematic picture of these two orbitals). This observation

aligns with Walsh diagram rules and with our intuition: forming a vacancy in the $3a_1$ orbital drives unbending motion in water, whereas a vacancy in the $1b_2$ orbital drives rapid dissociation of the deuterons[70,71]. Therefore, ionizing from the $3a_1$ orbital while keeping the $1b_2$ orbital intact allows substantial unbending prior to 3-body dissociation, permitting rapid slingshot motion. The largest contribution to the enhancement is from the $2^1A_1$ state, corresponding to a double vacancy in the $3a_1$ molecular orbital. Uniquely, 74% of all dissociations on this state undergo rapid slingshot motion, whereas this motion only occurs in 7% and 12% of dissociations on the other two states ($^3B_1$ and $^1B_1$ respectively).

**Modelling the EI mechanism**. We now attempt to model the enhancement mechanism that facilitates ionization of the third electron in $D_2O^{2+}$. Models of EI in diatomic cations such as $H_2^+$ often invoke a 1-dimensional tunnelling picture in which the double-well potential is distorted by a static field[64]. In this picture, the presence of the downhill hydrogen suppresses the tunneling barrier for electrons localized on the uphill hydrogen. The critical geometry is determined by balancing two competing factors: smaller bond lengths cause greater barrier suppression but larger bond lengths trap electronic population more effectively on the uphill hydrogen. Here, we will invoke a similar tunneling picture to explain EI in $D_2O^{2+}$ while noting the ways in which this case differs from the prototypical case of $H_2^+$.

The observed enhancement has a preferred polarization (Fig. 2h). All trications, whether formed by pulse pairs or single pulses, have their D-D axis parallel to the laser polarization axis ($\hat{\epsilon}$). For the pulse pairs, $\hat{\epsilon}$ refers to the polarization of the second pulse. In both the single and double pulse data, the alignment preference appears as a distribution of $D^+/D^+/O^+$ coincidences peaked at $\theta = 0$ and 180°. Utilizing the pulse pair data, the distribution in $\theta$ can be plotted as a function of delay, generating Fig. 5a. Here, the alignment preference is localized only around the enhancement at $\Delta t = 18$ fs. The distribution in $\theta$ becomes increasingly uniform far from this delay. A tunneling picture of ionization suggests that, for a brief window of time around $\Delta t = 18$ fs, the barrier to ionize is suppressed along $\theta = 0°$.

To construct our model of EI, we generated the molecular electrostatic potential (MEP) during the enhancement using the critical geometry discussed in the previous section. Modelling the

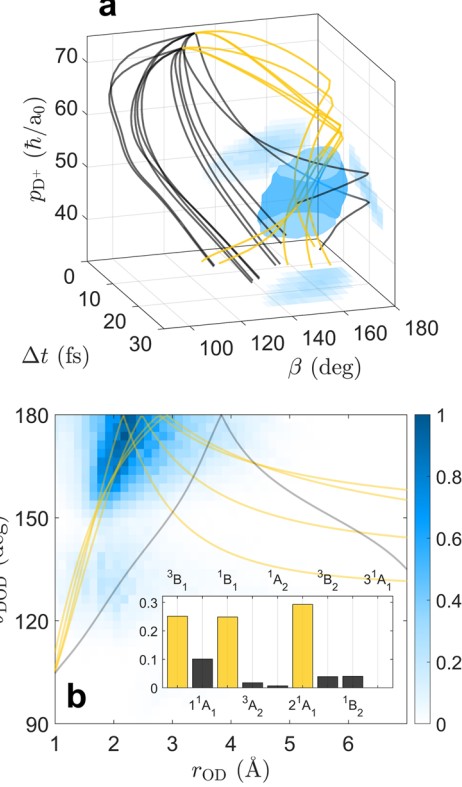

**Fig. 4 Localizing the enhancement in $D_2O^{2+} \rightarrow D_2O^{3+}$. a** A 3-dimensional representation (in $p_{D^+}$, $\beta$, and $\Delta t$) of the volume over which the enhanced trication production is within 50% of its maximum value (in semi-transparent cyan). 2-dimensional projections of the enhancement volume are displayed (in faded cyan) on the three walls of the plot. Two sample trajectories on each of the nine states of $D_2O^{2+}$ are also plotted within this 3-dimensional space. The trajectories on the states that undergo rapid slingshot motion are drawn in yellow, and those on the states that do not are drawn in black. **b** A 2-dimensional histogram of OD bond distance ($r_{OD}$) and bend angle ($\theta_{DOD}$) created by a weighted sum of the simulated trajectories. Five sample trajectories from (**a**) are reproduced in (**b**). The weighted state populations are shown in the inset of (**b**). Those depicted in yellow undergo rapid slingshot motion and those in black do not.

MEP for the ground state of $D_2O^{3+}$ at the approximate critical geometry ($r_{OD} = 2.2\text{Å}$ and $\theta_{DOD} = 180°$), subjecting it to our peak field strength ($\varepsilon_0 = 0.17\ E_h/ea_0$), and making two 1-dimensional cuts in $\theta$ ($\theta = 0°$ and $\theta = 90°$), yields Fig. 5b. If we incorporate the ionization potential (IP) of $D_2O^{2+}$ ($\sim 40.7\ \text{eV}$) in Fig. 5, the $\theta$ dependence of the ionization barrier becomes apparent (see Methods for details on the calculation of IP). When $\theta = 90°$, an electron localized near the oxygen atom must tunnel through a substantial barrier to ionize; however, when $\theta = 0°$, the barrier is nearly suppressed below the binding energy of the electron by the charge of the downhill deuteron, facilitating tunnelling (if not over-the-barrier ionization). This mechanism of barrier suppression is typical of EI phenomena previously observed in both diatomics[42–47,60–64] and linear triatomics[45,51,53,66]. Because the molecule is linear and stretched symmetrically, this same process may also occur on the opposite side of the molecule during the next half-cycle of the field. The third ionization most likely creates a vacancy in the $\sigma$ orbital of the linear molecule (the $1b_2$ orbital in $C_{2v}$ symmetry), pictured in Fig. 5b. This orbital, much like the other two valence orbitals, is predominantly localized at the oxygen atomic site when $r_{OD} = 2.2\text{Å}$, but unlike the other two valence orbitals, has the largest

value of electron density at the tunnelling barrier, and is doubly occupied in all three rapid slingshot states. The additional degeneracy introduced by unbending and symmetric stretching[70] may also supplement the tunneling current due to field-assisted couplings between states.

A distinct feature of EI in $H_2^+$ is enhancement at large ($> 5\ \text{Å}$) bond lengths. This preference has been widely attributed to the increased electron localization that occurs at large internuclear distances. Multiple phenomena contribute to this effect: one such phenomenon is known as charge resonance, whereby a unique property of the electronic states of stretched $H_2^+$ is exploited to localize electron density preferentially at one atomic site. EI that invokes this effect is known as Charge Resonance Enhanced Ionization (CREI)[60–64]. Another contributing phenomenon is electronic collision with the internal barrier of the double-well potential. Here, electronic population becomes trapped at a particular atomic site as the internal barrier of the double-well potential grows in height with increasing internuclear distance. We have evidence to suggest that these two phenomena do not play such prominent roles during EI in $D_2O^{2+}$.

As the ground electronic state of $H_2^+$ is stretched to infinity, the field-free charge distributions asymptotically become $H^+$/H and H/$H^+$, two "charge resonant" states that each localize electron density preferentially on a particular atomic site[72]. By contrast, when symmetrically stretching $D_2O^{2+}$, none of the nine field-free electronic states considered here continue to distribute electron density appreciably on either deuterium past an OD bond distance of $\sim 2\ \text{Å}$. As a result, the charge distribution of $D_2O^{2+}$ as a function of symmetric stretch in all 9 states becomes $D^+/D^+/O$ and notably not $D^+/D/O^+$[70]. This behavior not only precludes charge resonance at large internuclear distances, but additionally suggests that enhanced ionization in $D_2O^{2+}$ will occur at much smaller internuclear distances than in $H_2^+$, because stretching the molecule further than $\sim 2\ \text{Å}$ is not necessary to localize electron density on the desired atomic site.

In the proposed model of EI for $D_2O^{2+} \rightarrow D_2O^{3+}$, the global minimum of the tunneling barrier occurs at $r_{OD} \sim 1.8\ \text{Å}$. This distance is smaller still than the critical OD bond distance recovered in Fig. 4b ($r_{OD} = 2.2\ \text{Å}$), suggesting that the critical geometries extracted from our analysis may be more the result of the particular trajectories launched by double ionization, rather than representative of the global optimum in bond length for EI. This disparity does not occur in diatomic molecules because motion is only along one dimension; it is a feature in polyatomics due to the increased degrees of freedom: constraints exist for EI in both bend angle and bond length.

Figure 2f shows that single-pulses produce significantly greater KER than double pulses at the optimal delay. Single pulses with 19 or 40 fs duration produce a KER distribution peaked at 21.5 eV, while double pulses with an 18 fs delay have a KER distribution peaked at 17.5 eV. If the single pulse KER is simply due to Coulomb repulsion in the linear molecule, it implies a symmetric bond length of only 1.66 Å; however, this model excludes the kinetic energy acquired on the dication potential before the third ionization (see section 2.3 of the Supplementary Information including Fig. S3 for details). In the two-pulse experiment where this effect was modeled, the molecules undergoing slingshot motion have ~1.3 eV of kinetic energy just prior to the final ionization. We have no comparable field-dressed prediction for the single-pulse experiment, but we can assume that propagation on an intermediate potential adds some energy (1–2 eV) to the Coulomb explosion. Furthermore, the time-varying field may also distort the potentials due to phenomena such as bond softening[49,63]: one of the field-dressed states of the water dication, monocation, or both, could drive motion that

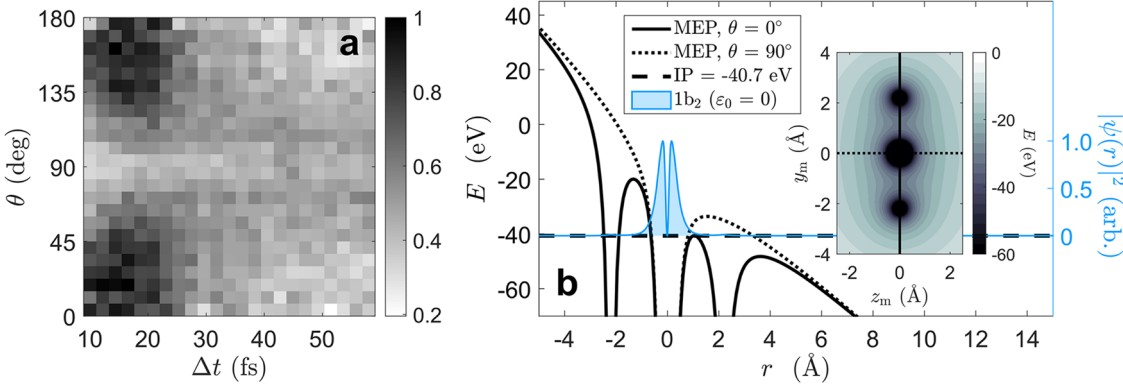

**Fig. 5 A model of enhanced ionization for $D_2O^{2+} \rightarrow D_2O^{3+}$. a** The distribution of alignment angle $\theta = \arccos(\hat{y}_m \cdot \hat{e})$ for all $D^+/D^+/O^+$ coincidences as a function of interpulse delay ($\Delta t$). **b** The simulated molecular electrostatic potential (MEP) of an unbent and stretched $D_2O^{3+}$ ($r_{OD} = 2.2$ Å and $\theta_{DOD} = 180°$) with a linearly polarized DC field strength of 0.17 $E_h/ea_0$ acting along two 1-dimensional cuts at $\theta = 0°$ (solid black line) and $\theta = 90°$ (dotted black line). Also plotted is the simulated ionization potential (IP) of $D_2O^{2+}$ (dashed black line) and the probability distribution $|\psi(r)|^2$ of the field-free ($\varepsilon_0 = 0$) $1b_2$ molecular orbital in the ground state of $D_2O^{2+}$ ($r_{OD} = 2.2$ Å and $\theta_{DOD} = 180°$). The inset to (**b**) displays a top-down view of the 2-dimensional field-free MEP including lines to represent both 1-dimensional cuts: $\theta = 0°$ in solid cyan and $\theta = 90°$ in dotted black.

unbends the molecule with less rapid stretching[73]. Evidence for propagation on field-dressed states is already visible in the alignment preference for longer pulse durations (Fig. 2h) where the degree of alignment for pulses 19 fs and longer is far greater than for shorter pulses. This suggests that dynamic alignment may have a substantial role in the EI process for long ($\tau \geq 19$ fs) pulses[54,55,57].

**Conclusions and outlook**. We have spatially and temporally resolved the rapid slingshot motion in $D_2O$ that induces EI within 20 fs. This motion is initiated by double ionization and creates, for a brief window in time, a linear DOD molecule with each OD bond distance stretched to about 2.2 Å. Within this window, a sufficiently strong and linearly polarized laser field oriented along the molecular axis can distort the molecular potential to facilitate ionization of a third electron and form $D_2O^{3+}$. Electrons localized near the oxygen atom see a suppressed tunneling barrier due to the charge of the downhill deuteron, and are thus more easily ionized. We see maximum EI when ionizing with 6 fs cross-polarized pulse pairs at pulse separations of 18 fs, but we also observe substantial EI when ionizing with single pulses of 19 fs duration or longer. In both cases EI appears to be associated with unbending and symmetric stretching of the molecular bonds, but the KER are different, suggesting additional field-dressed dynamics in the case of single-pulse EI.

Our measurements of the rapid intramolecular dynamics that produce EI in $D_2O^{2+}$ represent a new understanding of EI in polyatomics that can drastically affect the interpretation of CEI experiments. The largest limitation of CEI is often ultrafast motion of light atomic species in the intermediate sequence of charge states that are formed en route to Coulomb explosion. The distorting effects of EI are particularly egregious examples of this limitation. Understanding this motion is therefore required before CEI can be applied to more complex molecular systems.

This study shows not only how this motion can be measured in polyatomic molecules, but how the phenomenon of EI can be employed to do so. We demonstrated how EI can act as a structural filter to highlight a single kind of interatomic motion in water. It may be possible to tailor this EI process and select a subset of dynamics to highlight by changing the shape, intensity, or polarization of the ionizing laser field. This capability may have already been seen in recent experiments examining an elusive roaming reaction in formaldehyde[21].

Control of EI in polyatomic molecules could therefore extend the applicability of laser-induced CEI, and may spark a renewed interest in its use to film ultrafast motion within polyatomic molecules.

## Methods

**Producing pulses of variable duration**. In the first experimental scheme, single pulses of variable duration ($\tau = 6, 10, 19$, and 40 fs) and constant peak intensity ($I_0 = 2 \times 10^{15}$ W/cm$^2$) were used to triply ionize neutral $D_2O$. Here, $\tau$ is a measure of the full width at half maximum (FWHM) of the temporal intensity profile for each pulse. To create the 6 fs pulses, a 40 fs 800 nm 1 kHz Ti:sapphire laser pulse was spectrally broadened and chirped in a 1 m neon-filled hollow-core fiber (at 45 psi Ne) with a 250 $\mu$m diameter. The pulse (now with a central wavelength of 750 nm) was then recompressed using a series of bounces from two chirped mirror blocks. The 10 fs pulses were created similarly but using a lower gas pressure (35 psi Ne) to elicit less spectral broadening. The 19 fs pulses were created by positively chirping the 10 fs pulses using two fused silica wedges. Negatively chirping the 10 fs pulse to a duration of 19 fs yielded similar results. The 40 fs pulse was supplied directly by the output of a Ti:sapphire laser oscillator and amplifier. All peak intensities were kept constant by altering the energy per pulse using a series of neutral density pellicle filters with negligible dispersive effects. For the 6 and 10 fs pulses, temporal characterization was performed via the dispersion scan method utilizing the aforementioned fused silica wedges to apply variable dispersion[74]. For the longer pulse durations (19 and 40 fs), temporal characterization was performed via intensity autocorrelation[75].

**Producing pulse pairs with variable delay**. In the second experimental scheme, a pair of 6 fs 750 nm 1 kHz pulses with equal peak intensity ($I_0 = 1 \times 10^{15}$ W/cm$^2$) were used to triply ionize $D_2O$. These pulses were generated by first creating a single 6-fs pulse (as described in the section above) before splitting this pulse using a Mach-Zehnder interferometer. The interferometer splits the pulse into two pulses of equal intensity and variable delay. Each arm contained an additional polarizer at ± 45° to create a cross-polarized pair at the output directed along a common beam path. Pulse characterization was performed via dispersion scan utilizing two BK7 wedges to apply variable amounts of dispersion[74]. The interpulse delay was extracted with high precision from the spectral interference between the two beams in the unused output port of the interferometer.

**Detection geometry**. In either of the two experimental schemes (described by the two preceding sections), the laser was ultimately directed into a vacuum chamber with a pressure of $6 \times 10^{-10}$ Torr. The beam was then refocused back onto itself using a $f = 5$ cm in-vacuum spherical metal mirror to form a focal spot of ~7 $\mu$m. The chamber was then backfilled with a 50/50 mixture of gaseous $H_2O$ and $D_2O$ to a pressure of ~ $1.5 \times 10^{-9}$ Torr, such that < 1 molecule was in the focus during each laser shot on average. As depicted schematically in the center of Fig. 1a, the laser induced multiple ionization in $D_2O$, causing rapid distortions to the molecular geometry that ultimately result in a Coulomb explosion into molecular fragments. These fragments were then accelerated toward a detector by a series of electrostatic plates held at high voltage. The detector was comprised of a triple-stack of microchannel plates and a Roentdek delay-line hex-anode[68]. After post-processing of the electrical signals from the detector, this scheme yielded the full 3-dimensional momentum of each ionic fragment captured. With the laser

operating at a repetition rate of 1 kHz, we acquired all ions at an approximate count rate of ~ 500 counts/s or ~ 0.5 counts/shot.

**Recovering the molecular frame.** When captured in coincidence, fragment momenta are initially in lab-frame coordinates but can be rotated into an experimentally recovered molecular frame by defining a new set of coordinates: $x_m$, $y_m$ and $z_m$. Here, $\hat{z}_m$ is defined as the bisector of the two $D^+$ momenta, $\hat{x}_m$ as the cross product of the two $D^+$ momenta, and $\hat{y}_m$ as the cross product between $\hat{z}_m$ and $\hat{x}_m$:

$$\hat{z}_m = \left( \frac{\vec{p}_{D_{(1)}^+}}{|\vec{p}_{D_{(1)}^+}|} + \frac{\vec{p}_{D_{(2)}^+}}{|\vec{p}_{D_{(2)}^+}|} \right) / 2\cos(\beta/2) \quad (1a)$$

$$\hat{x}_m = \left( \vec{p}_{D_{(1)}^+} \times \vec{p}_{D_{(2)}^+} \right) / |\vec{p}_{D_{(1)}^+}||\vec{p}_{D_{(2)}^+}|\sin(\beta) \quad (1b)$$

$$\hat{y}_m = \hat{z}_m \times \hat{x}_m$$
$$\text{where } \beta = \arccos\left( \vec{p}_{D_{(1)}^+} \cdot \vec{p}_{D_{(2)}^+} \right) \quad (1c)$$

An important limitation of these coordinates is the lack of distinguishability between $+z_m$ and $-z_m$. Because the bisector of the two deuteron momenta is always defined as $+z_m$, there is nothing distinguishing a molecule that has been inverted about $z_m$ from one that has not. This inversion is only apparent after considering the evolution of certain time-resolved observables with pulse-pair separation. We see $\beta$ reach 180° before returning to more acute angles, indicating backward bending. We also see the magnitude of the oxygen-ion momentum ($p_{O^+}$) pass through zero, indicating a change of sign. (See section 1.1 of the Supplementary Information for further detail on the limitations of Eqs. (1a)–(1c).)

**Post-processing: eliminating false coincidence.** By focusing our analysis primarily on $D_2O$, rather than $H_2O$ or HOD, we were able to largely avoid false coincidences from the strong-field ionization of $H_2$, a prominent contaminant in gas-phase strong-field ionization experiments. Nonetheless, we employed a distinct technique in each channel of interest to further eliminate the contribution of false coincidence. For the coincidence channel of $D^+/D^+/O^+$, we made use of a momentum sum filter: all coincidences in which $|\vec{p}_{sum}|$ exceeded 25 $\hbar/a_0$ were discarded, where $|\vec{p}_{sum}|$ is defined as the magnitude of the vector sum over all particles' momenta ($|\vec{p}_{D_{(1)}^+} + \vec{p}_{D_{(2)}^+} + \vec{p}_{O^+}|$). For a real 3-body coincidence event, $|\vec{p}_{sum}|$ should ideally be zero, so this filtering process was extremely effective at eliminating the contributions of false coincidence. Notably, this same procedure could not be performed for the $D^+/D^+/O^{n+}$ channel as, in this channel, the $O^{n+}$ is not detected. In order to eliminate false coincidences in this channel, we applied a momentum filter in which all coincidences where $|\vec{p}_{D_{(i)}^+}|$ was below 10 $\hbar/a_0$ were discarded, where $|\vec{p}_{D_{(i)}^+}|$ is the magnitude of the momentum of either deuteron ($\vec{p}_{D_{(1)}^+}$ or $\vec{p}_{D_{(2)}^+}$). This filtering can be seen upon close inspection of Fig. 2a–e. The two remaining channels contributing to this coincidence ($D^+/D^+/O$ and $D^+/D^+/O^+$) were then easily distinguishable by the magnitudes of the deuteron momenta.

**Simulating the dynamics of $D_2O^{2+}$ and $D_2O^{3+}$.** To model the interatomic dynamics of $D_2O$ following double ionization, we simulated the motion of a nuclear wavepacket propagating on the dication potential energy surfaces semiclassically in the same way as was done in references[69] and[70], by using classical trajectories whose initial conditions are given by the Wigner phase space distribution of the initial vibrational state. The Wigner distribution in the harmonic approximation used here is

$$W(\mathbf{Q}, \mathbf{P}) = \frac{1}{(\pi\hbar)^3} \prod_{j=1}^{3} \exp\left[ -\frac{\omega_j}{\hbar} Q_j^2 - \frac{P_j^2}{\hbar\omega} \right], \quad (2)$$

where $Q_j$ and $P_j$ are the coordinates and momenta respectively of the three normal modes, and $\omega_j$ are the associated frequencies. The normal modes were calculated in a complete active space self-consistent field (CASSCF) calculation with the same active space used in the calculations of the dication potential surfaces described below. This semiclassical phase space distribution was propagated by sampling it to initiate a set of 2048 classical trajectories on each of the nine states of $D_2O^{2+}$. The vertical transition energies, symmetries, orbital vacancies in the dominant configuration at the equilibrium geometry, and branching ratios for all of these states are displayed in Table 1.

Eight of the three-dimensional potential energy surfaces used for this simulation were calculated by Gervais et al.[69] and the ninth, the $3^1A_1$ state, was calculated by Streeter et al.[70]. Briefly, the potentials were produced by internally contracted multi-reference configuration interaction (icMRCI) calculations at the configuration interaction singles and doubles (CISD) level including the Davidson correction for quadruple excitations. The calculations, which were performed on extensive grids of geometries, employed the cc-pVTZ Dunning correlation consistent basis[76] and were based on a CASSCF reference space. For example in the case of the $3^1A_1$ state, the active space used orbitals from calculations on the lowest $^3B_1$ state in $C_s$ symmetry with one $a'$ orbital frozen and six electrons in five $a'$ and

two $a''$ orbitals. These accurate ab initio surfaces calculated with MOLPRO[77,78] were fit to a linear combination of 100 basis functions that represent the Coulomb and polarization interactions at intermediate and long interatomic distances together with screened Coulomb and multipole interactions at short distances as described in reference[69].

The reliability of these potential surfaces and the Wigner phase space propagation on them for dissociative dynamics has been established in detailed comparisons with experimental momentum imaging experiments on one-photon double photoionization of $H_2O$ and $D_2O$[29,30,69,70]. This treatment of the nuclear dynamics closely reproduces experimental final momentum distributions for the three-body breakup channels as well as internal energy distributions in the two-body channels in those studies. Those benchmark comparisons underlie our confidence in the present investigation in using these detailed dynamics to interpret the experimental trajectories as described in this text.

After propagating on a given state of $D_2O^{2+}$ for a time commensurate with the interpulse delay, $\Delta t$, the formation of the $D_2O^{3+}$ and the resulting Coulomb explosion were modelled by an instantaneous transition (preserving the positions and momenta of the particles on the dication states) to a purely repulsive potential, $V$. This potential simply represents mutual Coulomb repulsion between three single charges:

$$V = \frac{q_1 q_2}{|\vec{r}_1 - \vec{r}_2|} + \frac{q_1 q_3}{|\vec{r}_1 - \vec{r}_3|} + \frac{q_2 q_3}{|\vec{r}_2 - \vec{r}_3|} \quad (3)$$

where $q_i$ is the net charge on each fragment ($q_i = 1$ in atomic units), $\vec{r}_i$ is the three-dimensional position vector of each fragment, and $i = 1, 2, 3$ correspond to $D_{(1)}^+$, $D_{(2)}^+$ and $O^+$ respectively. The inherent timing ambiguity due to the finite width of the two pulses in the pair ($\tau = 6$ fs) was accounted for by "blurring" the dication dynamics by ± 3 fs: that is, randomly shifting the timing of all 2048 trajectories forward and back by an amount between 0 and 3 fs and averaging over the results.

**Tunneling simulations.** The molecular electrostatic potential (MEP) for the ground quartet state of $D_2O^{3+}$ displayed in Fig. 5b (inset) was generated neglecting exchange interactions using restricted open-shell Hartree–Fock (ROHF) theory in GAMESS[79] with a 6-31G Gaussian basis set. A DC electric field of strength $\varepsilon_0 = 0.17$ $E_h/ea_0$ ($I_0 = 1 \times 10^{15}$ W/cm²) was then applied to this MEP along the $y_m$ and $z_m$ axes in order to yield the tunneling pictures in Fig. 5b for $\theta = 0$° and 90° respectively.

The probability distribution $|\psi(r)|^2$ of the field-free ($\varepsilon_0 = 0$) $1b_2$ molecular orbital for the triplet ground state of $D_2O^{2+}$ displayed in Fig. 5b was generated in the same way as the MEP: neglecting exchange interactions using ROHF theory in GAMESS with a 6-31G Gaussian basis set.

The ionization potential (IP) plotted in Fig. 5b as a horizontal line is from a CASSCF calculation of the difference between the energy of the ground state ($^3\Sigma^-$) of $H_2O^{2+}$ and that of $H_2O^{3+}$ ($^4\Sigma^-$) under the influence of an applied field of $\varepsilon_0 = 0.17$ $E_h/ea_0$. In these calculations, performed with the Psi4[80] suite of codes, the Gaussian basis was again the cc-pVTZ Dunning correlation consistent basis[76], and the OD distance was either 2 or 2.5 Å. Results were interpolated to yield the IP at 2.2 Å. The two tightest $a_1$ orbitals were doubly occupied in these calculations and the active space for calculations on both systems consisted of 2 $a_1$ 2 $b_1$ and 2 $b_2$ orbitals. Correlated calculations like this give ionization energies in the presence of an electric field several eV larger than ROHF descriptions. The difference evidently arises because CASSCF calculations can allow orbitals in correlating configurations to occupy the vicinity of the down-field deuteron, while single configuration-calculations spread the highest occupied orbital over both the oxygen and deuteron. As noted in the main text, these estimates of the ionization energy, and therefore the energy in a one-dimensional model at which tunneling or over-barrier ionization takes place, are very approximate. The calculations are complicated in particular by the difference in the electron correlation energies of the two systems, because the trication is dramatically less correlated than the dication. In addition, electronic states in intense fields have finite lifetimes and therefore energy widths, which are neglected in these calculations. Even if these computational estimates of the ionization energy and barrier height are off by one or two eV, the proposed mechanism for EI remains reasonable.

## Data availability
All data are available in the paper or the Supplementary Information.

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

## Acknowledgements
The authors thank M. Spanner for helpful discussion.

## Author contributions
A.J.H., M.B., and P.H.B. conceptualized the experiment. A.J.H., M.B., J.L.R., G.A.M., and I.G. conducted the experimental investigation. A.J.H., Z.L.S., and C.W.M. performed the theoretical investigation. A.J.H. performed formal analysis of the experimental data. A.J.H., M.B., R.F., G.A.M., and P.H.B. designed the experimental methodology. A.J.H., Z.L.S., R.R.L., and C.W.M. designed the theoretical methodology. A.J.H. and P.H.B. wrote the original draft of the paper. A.J.H., M.B., C.C., R.F., F.A., J.L.R., C.W.M., T.W., and P.H.B. reviewed and edited the paper.

## Funding
A.J.H., M.B., R.F., F.A., J.L.R., G.A.M., I.G., and P.H.B. were supported by the National Science Foundation. A.J.H. was additionally supported under a Stanford Graduate Fellowship as the 2019 Albion Walter Hewlett Fellow. R.F. acknowledges support from the Linac Coherent Light Source, SLAC National Accelerator Laboratory, which is supported by the US Department of Energy, Office of Science, Office of Basic Energy Sciences, under contract no. DE-AC02-76SF00515. I.G. was additionally supported by an NDSEG Fellowship. C.C. and T.W. were supported by the US Department of Energy under Award No. DE-FG02-08ER15984. Work at Lawrence Berkeley National Laboratory (LBNL) was performed under the auspices of the U.S. Department of Energy (DOE), Office of Science, Office of Basic Energy Sciences, Chemical Sciences, Geosciences, and Biosciences Division under Contract No. DE-AC02-05CH11231, using the National Energy Research Computing Center (NERSC), a DOE Office of Science User Facility, and the Lawrencium computational cluster resource provided by LBNL.

## Competing interests
The authors declare no competing interests.
