## [Peer review file · Communications Chemistry]

Reviewers' comments:

Reviewer #1 (Remarks to the Author):

Howard et al. present a detailed investigation of Coulomb explosion imaging (CEI) of D₂O molecules initiated with strong-field laser pulses. The paper concentrates on the dynamics of intermediate charge configurations, in particular in the D₂O(2⁺) state, and how these dynamics distort the CEI observables (kinetic energy release, dissociation bent angle and molecular frame alignment angle) in dependence of the duration of the ionizing laser pulse. In addition, the experiments reveal a very strong pulse-duration dependency of the tri- to di-cation yield ratio. These effects, which start to become apparent even for pulses below 10 fs, are analyzed by pump-probe double-pulse measurements with few-cycle pulses and comparison to ab initio simulations. Two particular trajectories (sling shot and two-body breakup) are found to be involved in the dicationic charge state, revealing detailed insight into the nuclear dynamics. Deeper analysis of the trajectories in terms of degree of enhanced ionization reveals that the slingshot trajectories are responsible for the strong dominance of the tricationic ionization channel within a time window of ~20 fs after population of the dicationic state, in agreement with the experiment.

I consider these results, which build on a number of strong-field papers published by the same group within the last five years, as novel and essential for the development of strong-field CEI as a probe for the nuclear structure of polyatomic molecules. The paper not only shows that dynamics of intermediate charge states (H₂O₂⁺), as a result of the sequential character of the strong-field ionization process, must be expected to blur the CEI observables, even for laser pulses in the range of 10-20 fs. With the analytical approach presented, the authors are able to identify and characterize the nuclear dynamics responsible for this distortion. I believe that these results are of considerable interest to other scientists in the field and will have implications on future studies.

The experimental results and simulations seem very reliable and the interpretation is sound. The paper is very well written. I therefore recommend publication of the paper in Communications Chemistry. Nevertheless, I have the following minor comments and questions:

1) Does nonaxial recoil play a role? If so, could it be that fast electronic relaxation processes in the dicationic state, resulting in a transient charge distribution, influence the observed time-varying momentum distributions in addition to the nuclear motions?

2) Can the authors comment on the relevance of sequential ionization with first one and then two electrons are removed, that is D₂O → D₂O⁺ → D₂O(3⁺)? In this case, dynamics in the singly ionized state D₂O⁺, as mentioned around line 121 in the paper, would influence the time-dependent momentum distributions of D⁺/O⁺/D⁺.

3) In line 148 ff the authors state: "1-dimensional distributions of KER, β, and θ for this particular interpulse delay are likewise reproduced in Figs. 2F–2H, also showing evidence for stretching, unbending, and alignment correlated with increased trication formation."

The molecular-frame alignment angle distribution for the double-pulse scheme shown in Figure 2H seems actually to be among the flattest curves shown in this plot. Probably, the authors could explain how they conclude on alignment for the double-pulse ionization.

4) In Figure 1B the value for R₀ should be specified with the same number of significant figures as the uncertainty, for example, R₀ = 0,4XY ± 0,006. Similar in Figure 1C and the figure caption.

5) Could the authors give a scale bar in Fig. 3G-P?

6) Line 228: Fig. 2B should read Fig. 3B.

7) Figure 4, caption: "Localizing the enhancement in D₂O₂⁺". Shouldn't this be D₂O₃⁺?

Reviewer #2 (Remarks to the Author):

The authors present a very interesting study of tracing atomic motion in a water molecule undergoing Coulomb explosion on the femtosecond timescale. The study combines advanced experimental and theoretical techniques to construct families of atomic trajectories that represent distinct dissociative ionization processes. The key process the paper focusses on is Enhanced Ionization (EI), a puzzling phenomenon in the early research with intense laser fields and still awaiting a consistent explanation across a wider range of molecules.

The paper is clearly written, with a significant effort to make the details accessible to non-specialists. I expect this work will be valuable not only to experts studying Coulomb explosion, but also to a wider community interested in the state-of-the-art experimental techniques of tracing ultrafast atomic motion. As far as I can judge, the work is original and significantly advances this field of research, striking the right balance between firm conclusions and tentative suppositions.

There are only a few, minor suggestions for improvement:

1. The trajectories plotted in the figures are crossing the quantum limits of classical physics. It would be helpful to highlight these limits, for example by plotting the Heisenberg uncertainty as ellipses on the momentum-time distributions, to warn the reader that the finer details of the trajectories are fundamentally uncertain.

2. The atomic unit abbreviation (a.u.) should be replaced by a combination of fundamental constants (\hbar/a_0 for momentum) as recommended by the International Union of Pure and Applied Chemistry. The reason for this recommendation is that it removes any ambiguity which unit "a.u." stands for and makes the conversion to the SI units straightforward.

3. The paper provides interesting evidence that charge resonance is not responsible for the EI process. There is another, widely used intuitive explanation of this process, namely electron collisions with the inner potential barriers at the onset of localization as the molecule stretches. A comment on whether such an explanation is applicable to this study would be useful.

4. The introduction suggests that strong-field studies of molecular dynamics started in 1995. In fact, our group pioneered this research 8 years earlier with diatomic molecules [L J Frasinski, et al, Phys. Rev. Lett. 58, 2424 (1987)] and followed these studies with triatomics [e.g. L J Frasinski, et al, Physics Letters A 156, 227 (1991)]. While we reported clear evidence for the fast EI process at that time, it is true that its mechanism was correctly explained only in 1995 [refs. 39, 40, and also J H Posthumus, et al, J. Phys. B 28, L349 (1995)].

I recommend accepting the manuscript for publication once the authors address the above points.

Leszek Frasinski, Imperial College London

Reviewer #3 (Remarks to the Author):

This paper reports experimental and theoretical studies on Coulomb explosion of D_2O , $D_2O^{3+} \rightarrow D^{+} + O^{+} + D^{+}$, by intense laser pulses ($1-2 \times 10^{15} \text{ W/cm}^2$). It is shown that the three-body explosion is enhanced by laser pulse stretching or by an additional ultrashort laser pulse introduced with a delay. The analysis of the molecular-frame momentum distributions indicates that the enhancement occurs at elongated molecular geometries near the linear configuration. The theoretical calculations show that the nuclear dynamics on electronic states of D_2O^{2+} , including those involving "slingshot" motion, are responsible for the structural change to the elongated structure. The experiments are well designed and the experimental and theoretical results are carefully analyzed. Enhanced ionization in intense laser fields has been discussed for more than three decades for different molecules. The present study, identifying the route towards the enhancement geometries, certainly deepens the understanding of Coulomb explosion imaging of small molecules. I believe that

the paper deserves publication if the following points are properly addressed by the authors.

(1) The advantage of Coulomb explosion imaging (CEI) in detecting to light atomic species is discussed in the introduction. I suggest to refer to the CEI studies on isomerization reactions of acetylene associated with migration of hydrogen atoms to signify the importance of time-resolved CEI, A. Hishikawa et al. Phys. Rev. Lett. 99, 258302 (2007), H. Ibrahim et al. Nat. Comm. 5, 5422 (2012) and C. Burger et al. Faraday Discuss. 194, 495 (2016).

(2) Coincidence measurements carried out with a relatively low repetition rate lasers (~ 1 kHz) often suffer from the contaminations from accidental coincidence events. More details of the experimental conditions should be given, such as the typical event rates and the procedure of coincidence analysis, in particular for the three-body breakup from dication, where the O fragment is not detected.

(3) The ionization is enhanced significantly at $\Delta t = 18$ fs. The enhancement is attributed to the ionization to the 3B1 1B1, 21A1 electronic states of the dication by the pump pulse, which unbends the molecular structure. The trication to dication ratio R is almost twice as large as that at 10 fs as shown in Fig. 1C. Does this mean that population to these states dominates over the others, e.g., the ground state (1A1) of the dication?

(4) The experiment results are interpreted under the assumption that D_2O^{3+} has a purely Coulombic potential. I am not quite sure if this is appropriate at this level of comparison, as the previous studies imply that the cationic potential energy surfaces deviates from purely Coulombic ones (see for example, Ref.37).

(5) Figure 3 clearly illustrates that different dissociation pathways are involved in the three-body Coulomb explosion. To complete the picture, the trajectories of the counter part D^+ ion of the two-body pathway should be plotted in Fig.3D.

(6) On page 6, the authors discuss that the charge resonance enhanced ionization (CREI) does not play a role in the enhancement in the dication, because none of dicationic states considered produce O^+ . If I understand correctly, CREI occurs via laser induced coupling between two charge transfer states having a large transition moment (see Ref. 36). I do not understand why this mechanism is excluded by the fact that O^+ is not produced by the dissociation.

(7) The barrier suppression model for enhanced ionization by neighboring ions within a molecule has been proposed more than decades ago. The discussion on tunneling ionization with Fig. 5B seems to give an impression that the model is proposed for the first time by this study. The authors should discuss the what is new here by referring to the previous studies, for example, Posthumus et al J. Phys. B28, L349 (1995), Ref. 39 and the extension to triatomic molecules, Brichta et al J. Phys. B40, 117 (2007).

Statement on the Revision of COMMSCHEM-22-0580-T in Response to Reviewer Feedback

A. J. Howard *et al.*

March 8, 2023

This statement addresses our recent revisions to the manuscript entitled “*Filming Enhanced Ionization in an Ultrafast Triatomic Slingshot*” in response to reviewer feedback. We would like to begin by thanking the editor for their careful consideration of this manuscript. We would also like to thank each of the three reviewers, not only for their recommendations of publication, but for their constructive criticism, and for their time spent thoroughly reviewing this work. It is our belief that the revisions detailed below have significantly strengthened the manuscript.

As for the formatting of this response letter: each Reviewer comment that we have directly addressed is reproduced below in a block of monospaced font followed by our response in no special formatting, and a reproduction of the relevant section of manuscript text in italics, with additions highlighted in green and deletions highlighted in red. Note: all references to line numbers refer to those in the original (unrevised) manuscript.

Comments by Reviewer #1

I consider these results, which build on a number of strong-field papers published by the same group within the last five years, as novel and essential for the development of strong-field CEI as a probe for the nuclear structure of polyatomic molecules. The paper not only shows that dynamics of intermediate charge states (H_2O^{2++}), as a result of the sequential character of the strong-field ionization process, must be expected to blur the CEI observables, even for laser pulses in the range of 10-20 fs. With the analytical approach presented, the authors are able to identify and characterize the nuclear dynamics responsible for this distortion. I believe that these results are of considerable interest to other scientists in the field and will have implications on future studies. The experimental results and simulations seem very reliable and the interpretation is sound. The paper is very well written. I therefore recommend publication of the paper in *Communications Chemistry*.

We thank Reviewer #1 for their kind words and their endorsement as to the novelty and importance of this manuscript.

Nevertheless, I have the following minor comments and questions...

We have responded to all comments and questions in the sections below.

1) Does nonaxial recoil play a role? If so, could it be that fast electronic relaxation processes in the dicationic state, resulting in a transient charge distribution, influence the observed time-varying momentum distributions in addition to the nuclear motions?

The term “nonaxial recoil”, as mentioned by Reviewer #1, generally refers to any case in which the axial-recoil approximation fails. This approximation assumes that when a bond is cleaved in a molecule, the resulting dissociative

molecular fragment will recoil with a momentum vector that is parallel to that bond. The cases in which this approximation fails are also referred to as “axial recoil breakdown” [29, 63]. As applied to Coulomb Explosion Imaging, this approximation only holds in the absence of any bending of molecular bond-angles and provided that each dissociating fragment is only acted upon by a point-like charge distribution concentrated at the atomic site that it is dissociating away from. Evidently, neither of these two cases hold true for the nuclear dynamics of D_2O^{2+} described in this manuscript; rapid unbending motion occurs prior to dissociation and each dissociated fragment is acted upon by multiple nuclei as well as a diffusely distributed charge distribution of the remaining valence electrons. Nonaxial recoil therefore plays a large role in shaping the dissociation dynamics of D_2O^{2+} . For a clear example of this, one can look to the discrepancy between the evolution of β as seen in Fig. 3C, and θ_{DOD} as seen in Figs. 3G-P. If the axial recoil approximation held, these two quantities would be identical. This discrepancy is also highlighted in the text in lines 212-214. In the interest of making this connection more explicit for the reader, we have supplemented this text, as follows:

In Fig. 3C, β is centered at 180° after only 10 fs. Figs. 3H and 3M demonstrate that this corresponds to a θ_{DOD} of only 146° and 158° , respectively. This discrepancy between β and θ_{DOD} represents a clear example of what is known as nonaxial recoil or a breakdown of the axial recoil approximation [29, 63].

The second point raised by Reviewer #1 is an interesting one, as fast electronic relaxation processes that influence nuclear motion would constitute a breakdown of the Born–Oppenheimer (BO) approximation. All of the semi-classical wavepacket simulations described in this paper invoke this approximation, so considering the cases in which it fails is important. The BO approximation breaks down whenever two or more potential energy surfaces approach each other in energy, for example near a Conical Intersection (CI). One such CI is revealed by inspection of Table I: whereas most of the 9 electronic states of D_2O^{2+} are energetically well separated at the Franck-Condon point, there are two singlet states that are nearly equal (± 0.1 eV) in energy: the 1A_2 and the 2^1A_1 . However, the single photon double-ionization experiments performed on H_2O in Refs. [29, 63] suggest that this instance of BO breakdown does not appreciably change the final ion momenta during the three-body breakup of the dication (into $H^+/H^+/O$). In those studies, Reedy and Streeter *et al.* utilized the final state products of this three-body dissociation (the final electronic state of the neutral oxygen and the final momenta of the two protons) to distinguish between up to eight different states of the dication (see Fig. 9 in Ref. [63]). In this analysis, the final experimental ion momenta of the protons dissociating from the 1A_2 and the 2^1A_1 states were both well approximated by semi-classical theoretical methods in which the BO approximation was applied. Although the final ion momentum distributions of this three-body dissociation differ for the cases of strong field ionization and single photon ionization, these differences were considered in great detail in Ref. [51], which concluded that couplings between dicationic states resulting from BO breakdown were not responsible.

2) Can the authors comment on the relevance of sequential ionization with first one and then two electrons are removed, that is $D_2O \rightarrow D_2O^+ \rightarrow D_2O^{(3+)}$? In this case, dynamics in the singly ionized state D_2O^+ , as mentioned around line 121 in the paper, would influence the time-dependent momentum distributions of $D^+/O^+/D^+$.

To be clear, lines 118-123, as mentioned by Reviewer #1, refer to the pathway: $D_2O \rightarrow D_2O^+ \rightarrow D_2O^{2+}$, rather than $D_2O \rightarrow D_2O^+ \rightarrow D_2O^{3+}$. The latter pathway is still possible, although none of the three lowest-lying states of the monocation (corresponding to ionization from the $1b_1$, $3a_1$, or $1b_2$ molecular orbitals) have an equilibrium geometry in which the molecule is simultaneously unbent and stretched to the same extent as in the dication [73]. In particular, among these three states, only one has an unbent equilibrium geometry (the 2^1A_1 ($3a_1$) $^{-1}$) however the equilibrium OD bond length in this state is only ~ 1 Å. As compared to dynamics in the dication, dynamics on this particular monocationic state would manifest as a channel in which the momentum of the deuterons remained comparatively higher and the unbending occurred more slowly. We do not see much evidence for this channel in, for example, Figs. 2A-C. We agree, however, that dynamics in the monocation en route to enhanced ionization are a great candidate for further study, especially considering how they may contribute to the enhancement observed for longer ($\tau \geq 20$ fs) single pulses. In order to emphasize this, we have supplemented the following text (around line 368):

“Fig. 2F shows that single-pulses produce significantly greater KER than double pulses at the optimal delay...the time-varying field may also distort the potentials due to phenomena such as bond softening (45, 56): one of the field-dressed states of the water dication, monocation, or both, for example, could drive motion that unbends the molecule with less rapid stretching.”

3) In line 148 ff the authors state: "1-dimensional distributions of KER, β , and θ for this particular interpulse delay are likewise reproduced in Figs. 2F-2H, also showing evidence for stretching, unbending, and alignment correlated with increased trication formation." The molecular-frame alignment angle distribution for the double-pulse scheme shown in Figure 2H seems actually to be among the flattest curves shown in this plot. Probably, the authors could explain how they conclude on alignment for the double-pulse ionization.

As noted by Reviewer #1, Fig. 2H reveals a stark contrast between the alignment distribution measured using long single pulses ($\tau \geq 20$ fs) and 6-fs pulse pairs (at $\Delta t = 18$ fs). In comparing these cases, it is clear that the alignment distribution is much more sharply peaked at $\theta = 0/180^\circ$ when multiply ionizing with long single pulses. We can attribute this to both the higher peak intensity of the single pulses (2×10^{15} W/cm² versus 1×10^{15} W/cm²), as well as the continued influence of the strong field over tens of femtoseconds, i.e. dynamic alignment [50]. For this reason, the alignment distribution acquired with 6-fs pulse pairs (the black curve in Fig. 2H) looks comparatively flat when plotted on the same scale as the alignment distributions acquired with long single pulses (the red and green curves in Fig. 2H). In fact, plotting these curves on the same scale was a purposeful choice to emphasize this difference. The minuscule size of the error bars on the black curve in Fig. 2H, however, indicates that this distribution deviates statistically significantly away from uniformity. This can be seen much more clearly by looking to Fig. 5A, which reproduces the black curve in Fig. 2H but plotted (in two dimensions) over all values of Δt rather than just $\Delta t = 18$ fs. Seen here, it is very clear that, whereas the distribution in θ may be considered roughly uniform at $\Delta t > 50$ fs, it is far from uniform at $\Delta t = 18$ fs. We agree, however, that we did not call sufficient attention to the difference between the distributions in θ for single pulses and pulse pairs in our discussion on lines 148-152, so we have added an extra sentence to clarify:

“1-dimensional distributions of KER, β , and θ for this particular interpulse delay are likewise reproduced in Figs. 2F-2H, also showing evidence for stretching, unbending, and alignment correlated with increased trication formation. The relative uniformity of the distribution in θ for $\Delta t = 18$ fs, as compared to the cases of $\tau = 20$ or 40 fs, however, indicates that dynamic alignment likely plays a much larger role in the nuclear dynamics en route to enhanced ionization when multiply ionizing with single long ($\tau \geq 20$ fs) pulses.”

4) In Figure 1B the value for R_0 should be specified with the same number of significant figures as the uncertainty, for example, $R_0 = 0,4XY \pm 0,006$. Similar in Figure 1C and the figure caption.

We agree with Reviewer #1 that R_0 should have been listed with the same precision as its uncertainty. We have made this correction in Figs. 1B and 1C, as well the figure caption, both of which are detailed below. We would like to thank Reviewer #1 for pointing this out.

... **(B)** The measured ratio (R) of triply-charged ($D^+/D^+/O^+$) to doubly-charged ($D^+/D^+/O$) three-body dissociations plotted logarithmically as a function of pulse duration (where $I_0 = 2 \times 10^{15} \text{ W/cm}^2$) and scaled to the ratio for a 6-fs pulse ($R_0 = 0.417 \pm 0.006$). **(C)** The same ratio as in **(B)** plotted as a function of interpulse delay (where $I_0 = 1 \times 10^{15} \text{ W/cm}^2$) and scaled to the ratio for a single 6-fs pulse ($R_0 = 0.039 \pm 0.003$). In both panels **(B)** and **(C)**, EI manifests as $R/R_0 > 1$.

5) Could the authors give a scale bar in Fig. 3G-P?

We originally included a scale bar for distances (representing 2 \AA) in Figs. 3G-P but we neglected to include an equivalent scale bar for momenta. We have since revised the figure to include a scale bar for momenta (representing $80 \hbar/a_0$), as seen below:

6) Line 228: Fig. 2B should read Fig. 3B.

Reviewer #1 is right. Corrected text is below:

“Likewise, in Fig. 2B3B, the faint cluster near $p_{O^+} \sim 80$ a.u. at $\Delta t \sim 60$ fs can also be attributed to this same motion in the two-body trajectory.

7) Figure 4, caption: “Localizing the enhancement in D2O2+”. Shouldn’t this be D2O3+?

As highlighted by Reviewer #1, there is an inherent ambiguity with this label: whether to name the initial or final state. In order to eliminate this ambiguity, we have changed the text in the caption of both Fig. 4 and Fig. 5 as follows:

“Figure 4: Localizing the enhancement in $D_2O^{2+}D_2O^{2+} \rightarrow D_2O^{3+}$ ”

“Figure 5: A model of enhanced ionization ~~in-for~~ $D_2O^{2+}D_2O^{2+} \rightarrow D_2O^{3+}$ ”

Comments by Reviewer #2

The authors present a very interesting study of tracing atomic motion in a water molecule undergoing Coulomb explosion on the femtosecond timescale. The study combines advanced experimental and theoretical techniques to construct families of atomic trajectories that represent distinct dissociative ionization processes. The key process the paper focusses on is Enhanced Ionization (EI), a puzzling phenomenon in the early research with intense laser fields and still awaiting a consistent explanation across a wider range of molecules.

The paper is clearly written, with a significant effort to make the details accessible to non-specialists. I expect this work will be valuable not only to experts studying Coulomb explosion, but also to a wider community interested in the state-of-the-art experimental techniques of tracing ultrafast atomic motion. As far as I can judge, the work is original and significantly advances this field of research, striking the right balance between firm conclusions and tentative suppositions.

We thank Reviewer #2 for their kind words and their compliments as to the writing and accessibility of this work.

There are only a few, minor suggestions for improvement...

We have addressed each suggestion in order below.

1. The trajectories plotted in the figures are crossing the quantum limits of classical physics. It would be helpful to highlight these limits, for example by plotting the Heisenberg uncertainty as ellipses on the momentum-time distributions, to warn the reader that the finer details of the trajectories are fundamentally uncertain.

We agree with Reviewer #2 that, taken alone, any one classical trajectory representing the nuclear dynamics in D_2O^{2+} technically violates the Heisenberg uncertainty principle as the energy, timing, position, and momentum of the nuclei are all simultaneously definite. However, taken together, the ensemble of trajectories that was propagated on

each state of the dication (2048 individual trajectories in each ensemble) represents an evolving Wigner phase-space distribution. The evolution of this distribution captures not only the inherent uncertainty of these quantities, but also the dispersion of the initial wave packet. Furthermore, this semi-classical method of sampling quantum nuclear dynamics with classical trajectories is a well established practice and similar methods have been used in many recent papers on the nuclear dynamics in the water dication [62, 63]. For these reasons, we believe the current representation is the most suitable means of presenting these simulations. We fear that drawing an ellipse to represent the Heisenberg uncertainty principle of each individual trajectory may confuse the reader.

2. The atomic unit abbreviation (a.u.) should be replaced by a combination of fundamental constants (\hbar/a_0 for momentum) as recommended by the International Union of Pure and Applied Chemistry. The reason for this recommendation is that it removes any ambiguity which unit "a.u." stands for and makes the conversion to the SI units straightforward.

We entirely agree with this recommendation from Reviewer #2, and we have since replaced every occurrence of "a.u." (that previously referred to atomic units of momentum) with " \hbar/a_0 " in both the figures and text. One such example of each is included below (lines 228-229 and Figs. 2A-2E):

Likewise, in Fig. 3B, the faint cluster near $p_{O^+} \sim 80$ a.u. $\cdot \hbar/a_0$ at $\Delta t \sim 60$ fs can also be attributed to this same motion in the two-body trajectory.

Similarly, we have also replaced all instances of "a.u." that refer to atomic units of electric field with E_h/ea_0 .

3. The paper provides interesting evidence that charge resonance is *not* responsible for the EI process. There is another, widely used intuitive explanation of this process, namely electron collisions with the inner potential barriers at the onset of localization as the molecule stretches. A comment on whether such an explanation is applicable to this study would be useful.

We agree with Reviewer #2 that, even when ignoring the effects of charge resonance, "electron collisions with the internal barrier" of a double-well potential can serve to localize electron density at the uphill atomic site and therefore prepare the conditions for enhanced ionization. This effect is most prominent at larger internuclear distances where the internal barrier is comparatively higher than the outer barrier. For the conditions described in this experiment, the internal barrier is significantly higher than the outer barrier, as seen in Fig. 5B, so this effect may play a role, as mentioned on lines 288-298, reproduced below:

Models of EI in diatomic cations such as H_2^+ often invoke a 1-dimensional tunnelling picture in which the double-well potential is distorted by a static field [57]. In this picture, the presence of the downhill hydrogen suppresses the tunneling barrier for electrons localized on the uphill hydrogen. The critical geometry is determined by balancing two competing factors: smaller bond lengths ~~will~~ cause greater barrier suppression but larger bond lengths ~~will~~ trap electronic population more effectively on the uphill hydrogen.

However, we would like to emphasize that this case differs from the prototypical case of enhanced ionization in H_2^+ . In the absence of an applied field, the ground state of H_2^+ distributes electron density equally on both atomic sites. Due to this initial charge distribution, many proposed mechanisms of enhanced ionization in H_2^+ first invoke a method by which electron density can become localized preferentially on the uphill atomic site prior to ionization. As the molecule is stretched to infinity, for instance, the charge distributions of H_2^+ asymptotically become H^+/H and H/H^+ , with electron density preferentially localized on either site. By contrast, none of the 9 electronic states of D_2O^{2+} here considered (in the absence of an applied field) continue to distribute electron density appreciably on the deuterium atomic site as the molecule is symmetrically stretched past an OD bond distance of $\sim 2 \text{ \AA}$. (As a result, the charge distribution as a function of symmetric stretch in all 9 states becomes $D^+/D^+/O$ and notably not $D^+/D/O^+$.) Therefore it may be that enhanced ionization occurs in D_2O^{2+} at much smaller internuclear distances than in H_2^+ because stretching the molecule further than $\sim 2 \text{ \AA}$ is not necessary to localize electron density on the desired atomic site.

We do not believe this particular line of argumentation was sufficiently clear in the original manuscript, though, leading to confusion from not only Reviewer #2 but also Reviewer #3. We have attempted to eliminate this confusion by expanding the arguments made starting on line 299, as seen below:

...Here, we will invoke a similar tunneling picture to explain EI in D_2O^{2+} while noting the ways in which this case differs from the prototypical case of H_2^+ .

~~A distinct feature of EI in H_2^+ is enhancement at large bond lengths ($>5 \text{ \AA}$) due to a phenomenon known as Charge Resonance Enhanced Ionization (CREI) [53-57]. We have evidence that this charge resonance effect does not play a similar role in the enhancement in D_2O^{2+} . Symmetric stretch in D_2O^{2+} yields the asymptotic products $D^+/D^+/O$ for all of the nine dicationic states considered here [63]. None of these states dissociate to O^+ as would be required for charge resonance effects to emerge [40,65]...~~

...The additional degeneracy introduced by unbending and symmetric stretching [63] may also supplement the tunneling current due to field-assisted couplings between states.

A distinct feature of EI in H_2^+ is enhancement at large ($>5 \text{ \AA}$) bond lengths. This preference has been widely attributed to the increased electron localization that occurs at large internuclear distances. Multiple phenomena contribute to this effect: one such phenomenon is known as charge resonance, whereby a unique property of the electronic states of stretched H_2^+ is exploited to localize electron density preferentially at one atomic site. EI that invokes this effect is known as Charge Resonance Enhanced Ionization (CREI) [53-57]. Another contributing phenomenon is electronic collision with the internal barrier of the double-well potential. Here, electronic population becomes trapped at a particular atomic site as the internal barrier of the double-well potential grows in height with increasing internuclear distance. We have evidence to suggest that these two phenomena do not play such prominent roles during EI in D_2O^{2+} .

As the ground electronic state of H_2^+ is stretched to infinity, the field-free charge distributions asymptotically become H^+/H and H/H^+ , two "charge resonant" states that each localize electron density preferentially on a particular atomic site [65]. By contrast, when symmetrically stretching D_2O^{2+} , none of the nine field-free electronic states considered here continue to distribute electron density appreciably on either deuterium past an OD bond distance of $\sim 2 \text{ \AA}$. As a result, the charge distribution of D_2O^{2+} as a function of symmetric stretch in all 9 states becomes $D^+/D^+/O$ and notably not $D^+/D/O^+$ [63]. This behavior not only precludes charge resonance at large internuclear distances, but additionally suggests that enhanced ionization in D_2O^{2+} will occur at much smaller internuclear distances than in H_2^+ , because stretching the molecule further than $\sim 2 \text{ \AA}$ is not necessary to localize electron density on the desired atomic site.

~~In this proposed model of EI for $D_2O^{2+} \rightarrow D_2O^{3+}$, the global minimum of the tunneling barrier occurs at $r_{OD} \sim 1.8 \text{ \AA}$. , notably different than the critical geometry indicated by Fig. 4B. This distance is smaller still than the critical OD bond distance recovered in Fig. 4B ($r_{OD} = 2.2 \text{ \AA}$), suggesting that the critical geometries extracted from our analysis This could indicate that the bond distances recovered in Fig. 4B are may be more likely the result of the particular trajectories launched by double ionization, rather than representative of the global optimum in bond length for EI. This disparity does not occur in diatomic molecules because motion is only along one dimension; , but it is a feature in polyatomics due to the increased degrees of freedom: constraints exist for EI in both bend angle and bond length.~~

In service of this point, we have also decided to include a 1-dimensional field-free probability distribution for the $1b_2$ molecular orbital (in the unbent and stretched ground state of D_2O^{2+}) in Fig. 5B. We believe that this visualization aptly demonstrates the extent to which electron density is localized at the oxygen atomic site after the molecule is stretched to an OD bond distance of 2.2 \AA . We have chosen to present this particular orbital, rather than the total electron density distribution, as we later mention that this is the orbital from which the third ionization event should predominantly take place. Revised figure and caption are below:

Figure 5: A model of enhanced ionization for $D_2O^{2+} \rightarrow D_2O^{3+}$. (A) The distribution of alignment angle $\theta = \arccos(\hat{y}_m \cdot \hat{\epsilon})$ for all $D^+/D^+/O^+$ coincidences as a function of interpulse delay (Δt). (B) The simulated molecular electrostatic potential (MEP) of an unbent and stretched D_2O^{3+} ($r_{OD} = 2.2 \text{ \AA}$ and $\theta_{DOD} = 180^\circ$) for—with a linearly polarized DC field strength of $0.17 E_h/ea_0$ acting along two 1-dimensional cuts at $\theta = 0^\circ$ (solid cyan-black line) and $\theta = 90^\circ$ (dotted black line). Also plotted is the simulated ionization potential (IP) of D_2O^{2+} (dash-dotted green-dashed black line)—and the probability distribution $|\psi(r)|^2$ of the field-free ($\epsilon_0 = 0$) $1b_2$ molecular orbital in the ground state of D_2O^{2+} ($r_{OD} = 2.2 \text{ \AA}$ and $\theta_{DOD} = 180^\circ$). The inset to panel (B) displays a top-down view of the 2-dimensional field-free MEP (prior to distortion by the DC field) including lines to represent both 1-dimensional cuts: $\theta = 0^\circ$ in solid cyan and $\theta = 90^\circ$ in dotted black.

The revised text associated with this figure is as follows:

The third ionization most likely creates a vacancy in the σ orbital of the linear molecule (the $1b_2$ orbital in C_{2v} symmetry), pictured in Fig. 5B. ~~as~~ This orbital, much like the other two valence orbitals, is predominantly localized at the oxygen atomic site when $r_{OD} = 2.2 \text{ \AA}$, but unlike the other two valence orbitals, has the largest value of electron density at the tunnelling barrier, and is doubly occupied in all three rapid slingshot states.

The corresponding text added to the Tunneling Simulations subsection of the METHODS section is below:

The molecular electrostatic potential (MEP) for the ground quartet state of D_2O^{3+} displayed in Fig. 5B (inset) was generated neglecting exchange interactions using restricted open-shell Hartree-Fock (ROHF) theory in GAMESS [71] with a 6-31G Gaussian basis set. A DC electric field of strength $\epsilon_0 = 0.17 \text{ a.u.}$ ($I_0 = 1 \times 10^{15} \text{ W/cm}^2$) was then applied to this MEP along the y_m and z_m axes in order to yield the tunneling pictures in Fig. 5B for $\theta = 0^\circ$ and 90° respectively.

The probability distribution $|\psi(r)|^2$ of the field-free ($\epsilon_0 = 0$) $1b_2$ molecular orbital for the triplet ground state of D_2O^{2+} displayed in Fig. 5B was generated in the same way as the MEP: neglecting exchange interactions using ROHF theory in GAMESS with a 6-31G Gaussian basis set.

4. The introduction suggests that strong-field studies of molecular dynamics started in 1995. In fact, our group pioneered this research 8 years earlier with diatomic molecules [L J Frasinski, et al, Phys. Rev. Lett. 58, 2424 (1987)] and followed these studies with triatomics [e.g. L J Frasinski, et al, Physics Letters A 156, 227 (1991)]. While we reported clear evidence for the fast EI process at that time, it is true that its mechanism was correctly explained only in 1995 [refs. 39, 40, and also J H Posthumus, et al, J. Phys. B 28, L349 (1995)].

We would like to thank Reviewer #2 for bringing these papers to our attention. We have since updated the introduction to include references to them:

This problem is greatly exacerbated by a strong-field phenomenon known as Enhanced Ionization (EI), where a “critical” spacing among the constituent atoms of a molecule increases the ionization yield [75, 39-43]...

...

Strong-field distortions of molecular dissociation dynamics were first studied extensively in diatomic molecules [39,74,75,40,44,45] and have been more recently studied in triatomic molecules [76,37,38,41,46]...

...

*[74] L. J. Frasinski, et al., Phys. Rev. Lett. **58**, 2424 (1987).*

*[75] J. H. Posthumus, et al. J. Phys. B **28**, L349 (1995).*

*[76] L. J. Frasinski, et al., Phys. Lett. A **156**, 227 (1991).*

Comments by Reviewer #3

The experiments are well designed and the experimental and theoretical results are carefully analyzed. Enhanced ionization in intense laser fields has been discussed for more than three decades for different molecules. The present study, identifying the route towards the enhancement geometries, certainly deepens the understanding of Coulomb explosion imaging of small molecules. I believe that the paper deserves publication if the following points are properly addressed by the authors.

We thank Reviewer #3 for their kind words and their acknowledgement of the impact of this work. We will address each point raised in order below.

(1) The advantage of Coulomb explosion imaging (CEI) in detecting to light atomic species is discussed in the introduction. I suggest to refer to the CEI studies on isomerization reactions of acetylene associated with migration of hydrogen atoms to signify the importance of time-resolved CEI, A. Hishikawa et al. Phys. Rev. Lett. 99, 258302 (2007), H. Ibrahim et al. Nat. Comm. 5, 5422 (2012) and C. Burger et al. Faraday Discuss. 194, 495 (2016).

We thank Reviewer #3 for bringing these papers to our attention. We have added these citations to the introduction, as below:

Many methods exist to record ultrafast motion in molecules including diffractive imaging techniques, such as ultrafast electron diffraction [7, 8, 9], laser-induced electron diffraction [10, 11], and hard x-ray diffraction [12, 13, 14]; spectroscopic techniques such as high harmonic generation [15, 16, 17]; and momentum imaging techniques such as Coulomb Explosion Imaging (CEI) [18-30,77-79] Diffractive techniques, however, lose sensitivity with decreasing atomic mass and spectroscopic techniques typically rely on indirect observables of atomic motion such as electronic structure. Among the methods listed, only CEI probes the direct momenta of all atoms, irrespective of mass, within a molecule [23,31,77-79] CEI deduces the positions of atoms within molecules by stripping away binding electrons and measuring the momenta of fragments produced in the resulting Coulomb repulsion of the ions.

...

*[77] A. Hishikawa, et al., Phys. Rev. Lett. **99**, 258302 (2007).*

*[78] H. Ibrahim et al., Nat. Comm. **5**, 4422 (2014).*

*[79] C. Burger et al., Faraday Discuss. **194**, 495 (2016).*

(2) Coincidence measurements carried out with a relatively low repetition rate lasers (~ 1 kHz) often suffer from the contaminations from accidental coincidence events. More details of the experimental conditions should be given, such as the typical event rates and the procedure of coincidence analysis, in particular for the three-body breakup from dication, where the O fragment is not detected.

We agree with Reviewer #3 that, for the sake of reproducibility, it would be helpful to include the average count rate during our data collection. We have added a sentence to the Detection Geometry discussion in the Methods section (lines 461-472) to do so:

“The chamber was then backfilled with a 50/50 mixture of gaseous H_2O and D_2O to a pressure of $\sim 1.5 \times 10^{-9}$ Torr, such that < 1 molecule was in the focus during each laser shot on average (~~operating at a repetition rate of 1 kHz~~)...After post-processing of the electrical signals from the detector, this scheme yielded the full 3-dimensional momentum of each ionic fragment captured. With the laser operating at a repetition rate of 1 kHz, we acquired all ions at an approximate count rate of ~ 500 counts/s or ~ 0.5 counts/shot.”

We also agree that a brief discussion on the post-processing procedure of the detected coincidences would be helpful to the reader. We have therefore added a sub-section to the METHODS section, entitled “Post-Processing: Eliminating False Coincidence”:

Post-Processing: Eliminating False Coincidence

By focusing our analysis primarily on D_2O , rather than H_2O or HOD , we were able to largely avoid false coincidences from the strong-field ionization of H_2 , a prominent contaminant in gas-phase strong-field ionization experiments. Nonetheless, we employed a distinct technique in each channel of interest to further eliminate the contribution of false coincidence. For the coincidence channel of $D^+/D^+/O^+$, we made use of a momentum sum filter: all coincidences in which $|\vec{p}_{\text{sum}}|$ exceeded $25 \hbar/a_0$ were discarded, where $|\vec{p}_{\text{sum}}|$ is defined as the magnitude of the vector sum over all particles' momenta ($|\vec{p}_{D(1)}^+ + \vec{p}_{D(2)}^+ + \vec{p}_{O^+}|$). For a real 3-body coincidence event, $|\vec{p}_{\text{sum}}|$ should ideally be zero, so this filtering process was extremely effective at eliminating the contributions of false coincidence. Notably, this same procedure could not be performed for the $D^+/D^+/O^{n+}$ channel as, in this channel, the O^{n+} is not detected. In order to eliminate false coincidences in this channel, we applied a momentum filter in which all coincidences where $|\vec{p}_{D(1)}^+|$ was below $10 \hbar/a_0$ were discarded, where $|\vec{p}_{D(1)}^+|$ is the magnitude of the momentum of either deuteron ($\vec{p}_{D(1)}^+$ or $\vec{p}_{D(2)}^+$). This filtering can be seen upon close inspection of Figs. 2A-E. The two remaining channels contributing to this coincidence ($D^+/D^+/O$ and $D^+/D^+/O^+$) were then easily distinguishable by the magnitudes of the deuteron momenta.

(3) The ionization is enhanced significantly at $\Delta t = 18$ fs. The enhancement is attributed to the ionization to the 3B_1 1B_1 , 2^1A_1 electronic states of the dication by the pump pulse, which unbends the molecular structure. The trication to dication ratio R is almost twice as large as that at 10 fs as shown in Fig. 1C. Does this mean that population to these states dominates over the others, e.g., the ground state (1A_1) of the dication?

We would first like to briefly clarify that, the triplet state (3B_1) is the energetic ground state of the water dication, rather than the singlet (1A_1), as mentioned by Reviewer #3. Nevertheless, Reviewer #3 brings to light an important point here, that is: although in Fig. 4B we highlight the three states that contribute most significantly to the observed enhancement, we do not claim to know if the first pulse in the pair actually populates these states more often than the others. This is an intentional omission by us, as we do not believe we have sufficient data to make such a claim, as discussed in the Supplemental Material, reproduced below:

Figs. S1G–I show that there is poor quantitative agreement between the data (Figs. S1A–S1C) and simulations (Figs. S1D–S1F), revealing regions in excess of 50% error. This is expected: the EI phenomenon is highly sensitive to the geometry as well as the occupied state of the molecule. The angular distribution in Fig. S1C, for example, has the most dense clustering of counts at $\Delta t \sim 18$ fs and $\beta \sim 150^\circ$ (corresponding to $\theta_{\text{DOD}} \sim 180^\circ$). The Wigner distribution of trajectories launched on any combination of the nine dication surfaces could never reproduce this clustering, because EI acts as a strong selective filter. The closest match comes from populating the 2^1A_1 state, in which (according to Table 1) 74% of the trajectories undergo the rapid “slingshot” motion that is favored by EI. The frequent occurrence of slingshot motion on the 2^1A_1 state explains why this analysis, as seen in Table S1, yields a maximal population in this state; however, the other states are still necessary to reproduce all the features seen in the experiment. For example, populating the higher lying states (such as $1B_2$) is necessary in order to reproduce the motion in which β unbends from 100° to 140° as Δt progresses from 10 to 30 fs. Ultimately, the lack of quantitative agreement between theory and experiment is a manifestation of the EI phenomenon. Furthermore, the subset of states that are promoted to the trication can vary as a function of delay; the state population distribution extracted for early delays yields different results than those extracted for late delays, further complicating any quantitative analysis of the state populations. For ~~all of~~ these reasons, the populations displayed in Table S1 should not be considered an accurate depiction of the initial ensemble of dicationic states launched by the initial pulse in the pair.

(4) The experiment results are interpreted under the assumption that D_2O^{3+} has a purely Coulombic potential. I am not quite sure if this is appropriate at this level of comparison, as the previous studies imply that the cationic potential energy surfaces deviates from purely Coulombic ones (see for example, Ref. 37).

The paper referenced by Reviewer #3 actually validates the use of Coulombic potentials for short (8 fs) pulses and high charge states (D_2O^{4+}), as quoted below:

“The term ‘Coulomb explosion’ implies that the Coulomb potential adequately approximates the true potential energy surface of the exploding ion when the charge state is high. We confirm the validity of this approximation by comparison with the *ab initio* potential for D_2O^{4+} ... The *ab initio* and Coulomb potentials yield nearly identical results for 8 fs laser pulses...we found no difference between the two potentials for molecular reconstruction in the 8 fs case.”

from [37] F. Légaré, *et al.*, *Phys. Rev. A* **71**, 013415 (2005).

Légaré *et al.* does, however, note that even shorter pulses may be able to better sample the difference between *ab initio* and Coulombic potentials for $D_2O \rightarrow D_2O^{4+}$ before the nuclear wavepacket has reached an area on the potential energy surfaces in which they converge. Since we are utilizing similar pulse durations to Légaré *et al.* (6 fs versus 8 fs), reaching similar charge states (D_2O^{3+} versus D_2O^{4+}), and recording data only after 10 fs of propagation on the dicationic potential energy surfaces ($\Delta t \geq 10$ fs), we believe the Coulombic approximation should be valid in our case.

(5) Figure 3 clearly illustrates that different dissociation pathways are involved in the three-body Coulomb explosion. To complete the picture, the trajectories of the counter part D^+ ion of the two-body pathway should be plotted in Fig.3D.

This is a great point, and we thank Reviewer #3 for bringing it to light. We have made the suggested addition to Figs. 3A and 3D. The updated figure and caption are below:

Figure 3: Filming nuclear motion in D_2O^{2+} The slingshot and two-body trajectories are highlighted in panels (D-F) ~~as a solid yellow and dashed black line, respectively~~ where a solid yellow line represents the slingshot trajectory, and the dashed (and dotted) black lines represent the two-body trajectory.

(6) On page 6, the authors discuss that the charge resonance enhanced ionization (CREI) does not play a role in the enhancement in the dication, because none of dicationic states considered produce O^{2+} . If I understand correctly, CREI occurs via laser induced coupling between two charge transfer states having a large transition moment (see Ref. 36). I do not understand why this mechanism is excluded by the fact that O^{2+} is not produced by the dissociation.

We believe this section of the manuscript was regrettably unclear in its original form, and we have since revised it in response to the feedback from both Reviewer #3 and Reviewer #2. Please see our response to the third comment by Reviewer #2 for context. The revised section is reproduced below:

...Here, we will invoke a similar tunneling picture to explain EI in D_2O^{2+} while noting the ways in which this case differs from the prototypical case of H_2^+ .

~~A distinct feature of EI in H_2^+ is enhancement at large bond lengths ($>5 \text{ \AA}$) due to a phenomenon known as Charge Resonance Enhanced Ionization (CREI) [53-57]. We have evidence that this charge resonance effect does not play a similar role in the enhancement in D_2O^{2+} . Symmetric stretch in D_2O^{2+} yields the asymptotic products $D^+/D^+/O$ for all of the nine dicationic states considered here [63]. None of these states dissociate to O^+ as would be required for charge resonance effects to emerge [40,65]...~~

...The additional degeneracy introduced by unbending and symmetric stretching [63] may also supplement the tunneling current due to field-assisted couplings between states.

A distinct feature of EI in H_2^+ is enhancement at large ($>5 \text{ \AA}$) bond lengths. This preference has been widely attributed to the increased electron localization that occurs at large internuclear distances. Multiple phenomena contribute to this effect: one such phenomenon is known as charge resonance, whereby a unique property of the electronic states of stretched H_2^+ is exploited to localize electron density preferentially at one atomic site. EI that invokes this effect is known as Charge Resonance Enhanced Ionization (CREI) [53-57]. Another contributing phenomenon is electronic collision with the internal barrier of the double-well potential. Here, electronic population becomes trapped at a particular atomic site as the internal barrier of the double-well potential grows in height with increasing internuclear distance. We have evidence to suggest that these two phenomena do not play such prominent roles during EI in D_2O^{2+} .

As the ground electronic state of H_2^+ is stretched to infinity, the field-free charge distributions asymptotically become H^+/H and H/H^+ , two "charge resonant" states that each localize electron density preferentially on a particular atomic site [65]. By contrast, when symmetrically stretching D_2O^{2+} , none of the nine field-free electronic states considered here continue to distribute electron density appreciably on either deuterium past an OD bond distance of $\sim 2 \text{ \AA}$. As a result, the charge distribution of D_2O^{2+} as a function of symmetric stretch in all 9 states becomes $D^+/D^+/O$ and notably not $D^+/D/O^+$ [63]. This behavior not only precludes charge resonance at large internuclear distances, but additionally suggests that enhanced ionization in D_2O^{2+} will occur at much smaller internuclear distances than in H_2^+ , because stretching the molecule further than $\sim 2 \text{ \AA}$ is not necessary to localize electron density on the desired atomic site.

In ~~this~~ the proposed model of EI for $D_2O^{2+} \rightarrow D_2O^{3+}$, the global minimum of the tunneling barrier occurs at $r_{OD} \sim 1.8 \text{ \AA}$. ~~, notably different than the critical geometry indicated by Fig. 4B. This distance is smaller still than the critical OD bond distance recovered in Fig. 4B ($r_{OD} = 2.2 \text{ \AA}$), suggesting that the critical geometries extracted from our analysis. This could indicate that the bond distances recovered in Fig. 4B are may be more likely~~ the result of the particular trajectories launched by double ionization, rather than representative of the global optimum in bond length for EI. This disparity does not occur in diatomic molecules because motion is only along one dimension; ~~, but~~ it is a feature in polyatomics due to the increased degrees of freedom: constraints exist for EI in both bend angle and bond length.

(7) The barrier suppression model for enhanced ionization by neighboring ions within a molecule has been proposed more than decades ago. The discussion on tunneling ionization with Fig. 5B seems to give an impression that the model is proposed for the first time by this study. The authors should discuss the what is new here by referring to the previous studies, for example, Posthumus et al J. Phys. B28, L349 (1995), Ref. 39 and the extension to triatomic molecules, Brichta et al J. Phys. B40, 117 (2007).

We thank Reviewer #3 for this feedback. In no way did we mean to imply we were inventing a new model of enhanced ionization in this paper. It was, however, important to us that we emphasize the key differences with the case of H_2^+ . After our response to the previous comment, we believe we have done so. In order to better address the lack of sufficient references to previous studies, we have supplemented the following text (starting line 329):

When $\theta = 90^\circ$, an electron localized near the oxygen atom must tunnel through a substantial barrier to ionize; however, when $\theta = 0^\circ$, the barrier is nearly suppressed below the binding energy of the electron by the charge of the downhill deuteron, facilitating tunnelling (if not over-the-barrier ionization). *This mechanism of barrier suppression is typical of EI phenomena previously observed in both diatomics [75,39-43,53-57] and linear triatomics [76,41,59].*

References

- [1] T. Brixner, *et al.*, *Nature* **434**, 625 (2005).
- [2] N. K. Schwab, F. Temps, *Science* **322**, 243 (2008).
- [3] V. I. Prokhorenko, *et al.*, *Science* **313**, 1257 (2006).
- [4] J. R. Dwyer, *et al.*, *Phil. Trans. R. Soc. A* **364**, 741 (2006).
- [5] A. Barty, J. Küpper, H. N. Chapman, *Annu. Rev. Phys. Chem.* **64**, 415 (2013).
- [6] M. Ivanov, *Faraday Discuss.* **228**, 622 (2021).
- [7] J. Yang, *et al.*, *Science* **361**, 64 (2018).
- [8] T. J. A. Wolf, *et al.*, *Nat. Chem.* **11**, 504 (2019).
- [9] J. Yang, *et al.*, *Science* **368**, 885 (2020).
- [10] M. Meckel, *et al.*, *Science* **320**, 1478 (2008).
- [11] B. Wolter, *et al.*, *Science* **354**, 308 (2016).
- [12] M. Minitti, *et al.*, *Phys. Rev. Lett.* **114**, 255501 (2015).
- [13] K. H. Kim, *et al.*, *Nature* **518**, 385 (2015).
- [14] J. Glowia, *et al.*, *Phys. Rev. Lett.* **117**, 153003 (2016).
- [15] W. Li, *et al.*, *Science* **322**, 1207 (2008).
- [16] H. J. Wörner, *et al.*, *Science* **334**, 208 (2011).
- [17] L. He, *et al.*, *Nat. Commun.* **9**, 1108 (2018).
- [18] M. Pitzer, *et al.*, *Science* **341**, 1096 (2013).
- [19] M. Kunitski, *et al.*, *Science* **348**, 551 (2015).
- [20] K. Fehre, *et al.*, *Sci. Adv.* **5**, eaau7923 (2019).
- [21] T. Endo, *et al.*, *Science* **370**, 1072 (2020).
- [22] B. Erk, *et al.*, *Science* **345**, 288 (2014).
- [23] C. E. Liekhus-Schmaltz, *et al.*, *Nat. Commun.* **6**, 8199 (2015).
- [24] S. Zeller, *et al.*, *Proc. Natl. Acad. Sci. U.S.A.* **113**, 14651 (2016).
- [25] A. Rudenko, *et al.*, *Nature* **546**, 129 (2017).
- [26] R. Boll, *et al.*, *Nat. Phys.* **18**, 423 (2022).
- [27] P. Herwig, *et al.*, *Science* **342**, 1084 (2013).
- [28] T. Jahnke, *et al.*, *Phys. Rev. X* **11**, 041044 (2021).
- [29] D. Reedy, *et al.*, *Phys. Rev. A* **98**, 053430 (2018).
- [30] T. Severt, *et al.*, *Nat. Commun.* **13**, 5146 (2022).
- [31] A. Matsuda, M. Fushitani, E. J. Takahashi, A. Hishikawa, *Multiphoton Processes and Attosecond Physics*, K. Yamanouchi, M. Katsumi, eds. (Springer Berlin Heidelberg, Berlin, Heidelberg, 2012), vol. 125, pp. 317–322.
- [32] Z. Vager, R. Naaman, E. P. Kanter, *Science* **244**, 426 (1989).
- [33] H. Stapelfeldt, E. Constant, P. B. Corkum, *Phys. Rev. Lett.* **74**, 3780 (1995).
- [34] H. Fukuzawa, *et al.*, *Nat. Commun.* **10**, 2186 (2019).
- [35] A. Rudenko, *et al.*, *Chem. Phys.* **329**, 193 (2006).
- [36] I. A. Bocharova, *et al.*, *Phys. Rev. A* **83**, 013417 (2011).
- [37] F. Légaré, *et al.*, *Phys. Rev. A* **71**, 013415 (2005).
- [38] F. Légaré, K. F. Lee, A. D. Bandrauk, D. M. Villeneuve, P. B. Corkum, *J. Phys. B* **39**, S503 (2006).
- [39] T. Seideman, M. Y. Ivanov, P. B. Corkum, *Phys. Rev. Lett.* **75**, 2819 (1995).
- [40] T. Zuo, A. D. Bandrauk, *Phys. Rev. A* **52**, R2511 (1995).
- [41] I. Bocharova, *et al.*, *Phys. Rev. Lett.* **107**, 063201 (2011).
- [42] H. Liu, *et al.*, *Chinese Physics Letters* **32**, 063301 (2015).
- [43] J. Wu, *et al.*, *Nat. Commun.* **3**, 1113 (2012).
- [44] H. Ibrahim, C. Lefebvre, A. D. Bandrauk, A. Staudte, F. Légaré, *J. Phys. B* **51**, 042002 (2018).
- [45] P. H. Bucksbaum, A. Zavriyev, H. G. Muller, D. W. Schumacher, *Phys. Rev. Lett.* **64**, 1883 (1990).

- [46] S. Zhao, *et al.*, *Phys. Rev. A* **99**, 053412 (2019).
- [47] G. A. McCracken, A. Kaldun, C. Liekhus-Schmaltz, P. H. Bucksbaum, *J. Chem. Phys.* **147**, 124308 (2017).
- [48] G. A. McCracken, P. H. Bucksbaum, *J. Chem. Phys.* **152**, 134308 (2020).
- [49] C. Cheng, *et al.*, *Phys. Rev. A* **102**, 052813 (2020).
- [50] A. J. Howard, *et al.*, *Phys. Rev. A* **103**, 043120 (2021).
- [51] C. Cheng, *et al.*, *Phys. Rev. A* **104**, 023108 (2021).
- [52] F. Allum, *et al.*, *J. Phys. Chem. Lett.* **12**, 8302 (2021).
- [53] C. Trump, H. Rottke, W. Sandner, *Phys. Rev. A* **59**, 2858 (1999).
- [54] C. Trump, H. Rottke, W. Sandner, *Phys. Rev. A* **60**, 3924 (1999).
- [55] T. Ergler, *et al.*, *Phys. Rev. Lett.* **95**, 093001 (2005).
- [56] I. Ben-Itzhak, *et al.*, *Phys. Rev. A* **78**, 063419 (2008).
- [57] H. Xu, F. He, D. Kielpinski, R. Sang, I. Litvinyuk, *Scientific Reports* **5**, 13527 (2015).
- [58] F. Légaré, *et al.*, *Phys. Rev. Lett.* **91**, 093002 (2003).
- [59] A. Matsuda, E. J. Takahashi, A. Hishikawa, *Journal of Electron Spectroscopy and Related Phenomena* **195**, 327 (2014).
- [60] S. Koh, K. Yamazaki, M. Kanno, H. Kono, K. Yamanouchi, *Chem. Phys. Lett.* **742**, 137165 (2020).
- [61] O. Jagutzki, *et al.*, *IEEE Transactions on Nuclear Science* **49**, 2477 (2002).
- [62] B. Gervais, *et al.*, *J. Chem. Phys.* **131**, 024302 (2009).
- [63] Z. L. Streeter, *et al.*, *Phys. Rev. A* **98**, 053429 (2018).
- [64] A. D. Walsh, *Journal of the Chemical Society (Resumed)* p. 2260 (1953).
- [65] R. S. Mulliken, *J. Chem. Phys.* **7**, 20 (1939).
- [66] M. Miranda, T. Fordell, C. Arnold, A. L’Huillier, H. Crespo, *Optics Express* **20**, 688 (2012).
- [67] J.-C. Diels, W. Rudolph, *Ultrashort laser pulse phenomena*, Optics and photonics (Elsevier / Academic Press, Amsterdam; Boston, 2006), second edn.
- [68] T. H. Dunning, *J. Chem. Phys.* **90**, 1007 (1989).
- [69] H.-J. Werner, P. J. Knowles, G. Knizia, F. R. Manby, M. Schütz, *WIREs Comput Mol Sci* **2**, 242 (2012).
- [70] H.-J. Werner, *et al.*, Molpro, version 2015.1, a package of ab initio programs (2015). See <http://www.molpro.net>.
- [71] G. M. J. Barca, *et al.*, *J. Chem. Phys.* **152**, 154102 (2020).
- [72] D. G. A. Smith, *et al.*, *J. Chem. Phys.* **152**, 184108 (2020).
- [73] C. F. Jackels, *J. Chem. Phys.* **72**, 4873 (1980).
- [74] L. J. Frasinski, *et al.*, *Phys. Rev. Lett.* **58**, 2424 (1987).
- [75] J. H. Posthumus, *et al.* *J. Phys. B* **28**, L349 (1995).
- [76] L. J. Frasinski, *et al.*, *Phys. Lett. A* **156**, 227 (1991).
- [77] A. Hishikawa, *et al.*, *Phys. Rev. Lett.* **99**, 258302 (2007).
- [78] H. Ibrahim *et al.*, *Nat. Comm.* **5**, 4422 (2014).
- [79] C. Burger *et al.*, *Faraday Discuss.* **194**, 495 (2016).
- [80] J. P. Brichta *et al.*, *J. Phys. B* **40**, 117 (2007).

REVIEWERS' COMMENTS:

Reviewer #1 (Remarks to the Author):

I thank the authors for their very detailed response to my and the other reviewers' comments and questions, and for the corresponding changes in the manuscript. All my concerns are resolved and I recommend publication of the paper as it is.

Reviewer #2 (Remarks to the Author):

I am pleased to see that the points raised in the previous round of review have been satisfactorily addressed. I recommend publishing the current version of the manuscript.

Leszek Frasiniski, Imperial College London

Reviewer #3 (Remarks to the Author):

The authors properly addressed all of the previous comments by the reviewers. I believe that the paper can be published in Comm. Chem. in the present form.